# A Novel Integrative Framework for Depression: Combining Network Pharmacology, Artificial Intelligence, and Multi-Omics with a Focus on the Microbiota–Gut–Brain Axis

**DOI:** 10.3390/cimb47121061

**Published:** 2025-12-18

**Authors:** Lele Zhang, Kai Chen, Shun Li, Shengjie Liu, Zhenjie Wang

**Affiliations:** 1School of Biological and Food Engineering, Fuyang Normal University, 100 Qinghe West Road, Fuyang 236037, China; zhanglele@fynu.edu.cn (L.Z.); 23211304@stu.fynu.edu.cn (S.L.); 2College of Information Engineering, Fuyang Normal University, 741 Qinghe East Road, Fuyang 236041, China; 19355695598@163.com; 3College of Food Science and Technology, Nanjing Agricultural University, No. 1 Weigang Road, Nanjing 210095, China

**Keywords:** depression, network pharmacology, artificial intelligence, multi-omics, microbiota–gut–brain axis

## Abstract

Major Depressive Disorder (MDD) poses a significant global health burden, characterized by a complex and heterogeneous pathophysiology insufficiently targeted by conventional single-treatment approaches. This review presents an integrative framework incorporating network pharmacology, artificial intelligence (AI), and multi-omics technologies to advance a systems-level understanding and management of MDD. Its central contribution lies in moving beyond reductionist methods by embracing a holistic perspective that accounts for dynamic interactions within biological networks. The primary objective is to demonstrate how AI-powered integration of multi-omics data—spanning genomics, proteomics, and metabolomics—can enable the construction of predictive network models. These models are designed to uncover fundamental disease mechanisms, identify clinically relevant biotypes, and reveal novel therapeutic targets tailored to specific pathological contexts. Methodologically, the review examines the microbiota–gut–brain (MGB) axis as an illustrative case study, detailing its pathogenic roles through neuroimmune alterations, metabolic dysfunction, and disrupted neuro-plasticity. Furthermore, we propose a translational roadmap that includes AI-assisted biomarker discovery, computational drug repurposing, and patient-specific “digital twin” models to advance precision psychiatry. Our analysis confirms that this integrated framework offers a coherent route toward mechanism-based personalized therapies and helps bridge the gap between computational biology and clinical practice. Nevertheless, important challenges remain, particularly pertaining to data heterogeneity, model interpretability, and clinical implementation. In conclusion, we stress that future success will require integrating prospective longitudinal multi-omics cohorts, high-resolution digital phenotyping, and ethically aligned, explainable AI (XAI) systems. These concerted efforts are essential to realize the full potential of precision psychiatry for MDD.

## 1. Introduction

Depression, a debilitating mental disorder, poses a significant global health burden [1], affecting approximately 12% of adults globally and contributing substantially to disability and suicide [2]. Despite its high prevalence, the etiopathogenesis of depression remains elusive, largely due to its inherent complexity, multifactorial nature, and profound clinical heterogeneity [3]. Patients present with a wide spectrum of symptoms, severity, and treatment responses, underscoring that depression is not a single disease entity but rather a common clinical endpoint arising from a variety of underlying biological disturbances.

Despite the availability of various classes of antidepressants—including the less commonly prescribed monoamine oxidase inhibitors, and the widely used selective serotonin reuptake inhibitors and serotonin-norepinephrine reuptake inhibitors—their clinical utility remains limited by a significant lag in efficacy [4,5], high rates of treatment resistance (approximately 30%) [3], and a multitude of adverse side effects that compromise patient adherence [6]. These limitations stem from a historical reductionist approach that targets single molecules or pathways, overlooking the polygenic and multisystemic nature of depression. This outdated “one-size-fits-all” paradigm underscores the critical unmet need for more precise strategies. Thus, the development of novel therapeutic agents remains a priority. This impasse necessitates a fundamental shift from a reductionist to a systems-level framework to deconvolute the disease’s complexity effectively.

The conceptualization of depression has evolved beyond neurotransmitter deficits to encompass dysregulations across multiple biological systems. Advances in genetics and systems biology have revealed that depression arises from the intricate interplay of polygenic risk scores [7], epigenetic modifications [8], dysregulated immune-inflammatory pathways [9], neuroendocrine (HPA axis) dysfunction [10], and alterations in gut–brain axis communication [11]. This complex network of interactions implies that effective intervention must move beyond isolated targets to modulate the entire disease-perturbed network. Thus, a systems-level understanding of depression requires a paradigm shift from a reductionist to a holistic perspective. Although multi-omics studies have catalogued molecular changes and network pharmacology has proposed new targets, these efforts have largely operated in isolation. The lack of integration between data generation and mechanistic interpretation remains a significant bottleneck, perpetuating the translational gap between discovery and effective therapeutics.

To address this challenge, an integrative framework combining network pharmacology, artificial Intelligence (AI), and multi-omics technologies emerges as a transformative, novel paradigm. Individually, each discipline offers unique and complementary capabilities: First, multi-omics (genomics, transcriptomics, proteomics, metabolomics) provides a comprehensive, multi-layer mapping of the molecular landscape underlying depression [12]. Building on this foundational data, Network Pharmacology constructs and analyzes the complex web of interactions between drugs, targets, and disease networks to predict therapeutic effects and identify multi-target interventions [13]. Finally, AI and Machine Learning (ML) are indispensable for integrating these massive, heterogeneous datasets, extracting meaningful patterns, and generating predictive, testable hypotheses [14]. Ultimately, this synergistic convergence—where AI dynamically interprets multi-omics data to inform and validate network pharmacology models—transforms these parallel tools into a unified, hypothesis-generating system.

This review aims to provide a comprehensive synthesis of this novel paradigm, outlining a pathway from data integration to therapeutic discovery. Throughout this review, the microbiota-gut–brain (MGB) axis is employed as a central, illustrative case study to demonstrate how each component of this paradigm—multi-omics, network pharmacology, and AI—can be integrated to decode a specific, complex pathophysiological system in depression. First, multi-omics profiling is detailed to unveil the molecular architecture of depression. This will be followed by a discussion of how network pharmacology models the mechanism of action of antidepressants and identifies potential multi-target interventions. Subsequently, the role of AI in integrating these data is highlighted for the discovery of diagnostic, prognostic, and treatment-response biomarkers, patient stratification (subtyping), and drug repurposing. The convergence of these approaches provides a more profound understanding of complex neuropsychiatric conditions and enhances the development of targeted treatment strategies, as illustrated in the integrative framework depicted in Figure 1. Finally, the review will conclude with a discussion of current challenges and future directions, proposing that this integrative approach is not merely an alternative but a necessity for decoding the heterogeneity of depression, thereby ushering in a new era of personalized molecular pharmacotherapeutics.

While the individual potential of multi-omics, network pharmacology, and AI in depression research has been recognized in existing literature, a critical synthesis gap remains. Previous reviews have often focused on one or two of these domains in isolation—for instance, discussing multi-omics biomarkers in depression [15], outlining the principles of network pharmacology [16], or exploring the role of the gut–brain axis [17]. However, there is a notable absence of comprehensive frameworks that explicitly integrate all three pillars—network pharmacology, AI, and multi-omics—into a cohesive paradigm to advance molecular therapeutics for depression.

To address these limitations, this review provides the first comprehensive synthesis. It proposes a unified, iterative framework in which AI serves as a central engine to dynamically interpret multi-omics data, thereby informing and validating context-aware network pharmacology models. The work makes three key contributions. First, it moves beyond isolated discussions to demonstrate the synergistic potential of integrating these three approaches. Second, it employs the MGB axis not merely as a topic of review but as a central, systems-level case study to concretely illustrate how the proposed framework can decode a specific, complex pathophysiological system—from multi-omics data integration and AI-driven pattern discovery to network-based target prioritization and therapeutic hypothesis generation. Finally, it outlines a clear translational pathway from large-scale data integration to novel diagnostic and therapeutic strategies, providing a strategic roadmap for researchers and clinicians in precision psychiatry. By bridging this methodological gap, this review aims to foster a new foundational perspective for future research in the field.

## 2. Methods

This systematic review aimed to identify all relevant studies on the integration of multi-omics, AI, and network pharmacology in depression research, with a specific focus on the MGB axis. To ensure the inclusion of the most current evidence, a comprehensive and reproducible search was conducted across three major electronic databases—PubMed/MEDLINE, Web of Science, and Google Scholar—with publication dates restricted to the past 5 years (from 2020 to 15 October 2025). The search utilized controlled terms with the following combinations: (“Depression” OR “Major Depressive Disorder”) AND (“Artificial Intelligence” OR “Machine Learning” OR “Deep Learning” OR “AI”); (“Depression” OR “Major Depressive Disorder”) AND (“Multi-Omics” OR “Metagenomics” OR “Proteomics” OR “Transcriptomics” OR “Metabolomics”); and (“Depression” OR “Major Depressive Disorder”) AND (“Microbiota-Gut-Brain Axis” OR “Gut-Brain Axis” OR “Gut Microbiota” OR “Microbiome”).

Following the search, all retrieved records were imported into EndNote for automatic and manual deduplication. Subsequently, study selection proceeded in two stages based on predefined eligibility criteria. The inclusion criteria encompassed original research or review articles focused on major depressive disorder in humans that explicitly integrated at least two of the three core domains (multi-omics, AI, network pharmacology) within the context of the MGB axis. Studies were excluded if they were not published in English, were conference abstracts, editorials, letters, or book chapters, or were based solely on animal or cell models without correlative human data. Titles and abstracts were screened first for relevance, followed by a full-text assessment of potentially eligible articles. To minimize selection bias, two reviewers independently conducted the entire screening process; any discrepancies were resolved through consensus or adjudication by a third reviewer.

Finally, data from the included studies were systematically extracted into a standardized form capturing key information, including authors, publication year, study design, types of omics data, specific AI/machine learning methods, network pharmacology approaches, key findings related to the MGB axis, and key conclusions for narrative synthesis.

## 3. AI-Powered Integration of Network Pharmacology and Multi-Omics

Traditional research paradigms have encountered significant limitations in deciphering the complexity of MDD. The failure of single-target strategies and the highly heterogeneous clinical presentations collectively indicate a fundamental need to shift from a static, reductionist perspective to a dynamic, systems-level one. This section builds upon these limitations to elaborate an integrative paradigm that converges network pharmacology, multi-omics, and AI. Here, AI acts not in isolation but as a central engine that unlocks the synergistic potential of the other two fields, collectively driving a methodological revolution in depression research. The core of this revolution lies in the deep integration of next-generation network pharmacology with multi-omics data, leveraging the computational power and advanced algorithms of AI to construct models that reflect the true complexity of the disease, ultimately enabling the seamless translation of big data into novel discoveries.

### 3.1. Multi-Omics Data Generation and Integration: A Foundation for Systems-Level Insight

Multi-omics technologies provide multifaceted and complementary biological insights into MDD. Their integration is fundamental for reconstructing causal disease networks—from genetic predisposition to physiological output—thereby advancing from associative findings to a mechanistic understanding of MDD.

#### 3.1.1. Genomics: Defining Inherited Risk

Genome-wide association studies (GWAS) have identified numerous genetic loci associated with MDD risk. For example, the landmark study by Howard et al., which included over 807,553 cases, identified 102 independent variants, implicated 269 genes, and highlighted 15 gene sets associated with depression. These findings underscore pathways related to neuronal development and synaptic function [18]. Polygenic risk scores derived from these findings quantify individual genetic susceptibility, providing a foundational layer for understanding disease heritability [19].

#### 3.1.2. Epigenomics: Bridging Environment and Genetics

Epigenetic mechanisms have emerged as crucial mediators connecting genetic predisposition and environmental exposures in the pathophysiology of depression. These mechanisms—including DNA methylation, hydroxymethylation, and histone modifications—produce stable alterations in gene expression without changing the underlying DNA sequence and have been consistently implicated in various psychiatric disorders [20]. Supporting this paradigm, a study by Efstathopoulos et al., which analyzed salivary DNA from a cohort of 1149 adolescents, demonstrated that methylation levels in exon 1 of the *NR3C1* gene mediated the association between early life stress and the severity of depressive symptoms [21]. Such findings directly illustrate how life experiences “program” gene expression patterns to influence disease pathology.

#### 3.1.3. Transcriptomics: Mapping Functional Gene Activity

RNA sequencing (RNA-seq) enables comprehensive profiling of global gene expression patterns in specific tissues or cell types under disease conditions, directly reflecting functional gene activity and serving as a primary resource for identifying key pathways and biomarkers [22]. Advancing beyond bulk tissue analysis, single-nucleus RNA sequencing (snRNA-seq) now enables the precise mapping of transcriptional alterations to specific cell populations, uncovering unprecedented cellular-resolution insights into disease mechanisms. Nagy et al. employed single-nucleus transcriptomics on the dorsolateral prefrontal cortex in MDD, revealing widespread, cell-type-specific gene expression changes. Specifically, deep-layer excitatory neurons and immature oligodendrocyte precursor cells were the most severely dysregulated, accounting for nearly half (47%) of all changes, directly implicating these populations in disease pathology and highlighting them for future investigative and therapeutic focus [23].

#### 3.1.4. Proteomics: From Biomarker Discovery to Personalized Therapy

Proteomics plays an essential role in identifying biomarkers and informing personalized treatment strategies [24]. This is exemplified by the large-scale study by Bot et al., which analyzed 1589 participants and identified several depression-associated proteins implicated in critical processes such as immune response and cell communication [25], thereby fulfilling the core aim of biomarker discovery. Furthermore, the dynamic nature of the proteome is highlighted by studies showing that many inflammatory proteins exhibit significant concentration changes following antidepressant treatment [26], directly linking proteomic profiles to treatment response.

#### 3.1.5. Metabolomics: Profiling Systemic Physiological Output

As the terminal downstream readout of physiological activity, metabolomics provides a unique window into the functional outputs of cellular processes and can reflect central nervous system activity across the blood–brain barrier [27]. Studies of young patients with MDD have revealed perturbations in specific metabolic pathways, including purine and fatty acid metabolism. Notably, inosine was identified as a potential independent diagnostic biomarker. Furthermore, the pathophysiology of MDD in this population appears to differ from that of adults, particularly in tryptophan metabolism [28]. The dysregulation of amino acid metabolism in MDD is characterized by elevated serum levels of glutamate, aspartate, and glycine, alongside decreased 3-hydroxykynurenine. Additionally, the serum levels of glutamate and phenylalanine show a correlation with depression severity, indicating their potential as clinical markers for severity-based stratification [29].

A summary of these multi-omics technologies and their contributions to MDD research is provided in Table 1.

#### 3.1.6. Integrative Multi-Omics: Towards a Unified Model

As highlighted in Table 1, integrating multiple omics layers is essential for reconstructing causal networks spanning from genetic variation to metabolic output. The individual omics layers, as detailed in the preceding sections, provide multifaceted but fragmented insights into MDD. Therefore, integrating these diverse data types through advanced computational methods is pivotal for constructing a unified systems-level model. A particularly promising frontier involves integrating host multi-omics data with gut metagenomic profiles, a fusion strategy that is critical for systematically elucidating the role of the MGB axis in MDD and, as detailed in later sections, paving the way for precision neuropsychiatry. This systemic approach is indispensable for dissecting disease heterogeneity, delineating molecular subtypes, and elucidating the mechanistic interplay between genetic predisposition, environmental triggers, and dysregulated biological pathways.

### 3.2. The Network Pharmacology Paradigm: From Single-Target to Systems-Level Intervention

#### 3.2.1. Fundamentals of Network Pharmacology: From Single-Target to Network Medicine

The traditional organ-focused medical paradigm and the reductionist “one disease–one target–one drug” approach have substantially limited advances in understanding complex diseases and developing effective therapeutics. In contrast, network pharmacology (NP) has emerged as a discipline that transcends this limitation by adopting a systems-level perspective. First conceptualized by Hopkins [30], NP integrates systems biology, computational biology, and experimental validation through a data-driven framework to holistically decipher complex compound–body interaction networks [31].

Guided by the concept of “network targeting,” NP offers innovative strategies to elucidate how multi-compound or drug combinations act synergistically on key network proteins, thereby modulating shared disease-associated modules or signaling pathways [32]. This enables a fundamental shift from symptomatic treatment to targeting causal mechanisms in complex diseases [33].

Central to NP methodology is the construction and analysis of a “drug–target–disease” heterogeneous network, which integrates multi-source data, including protein–protein interactions to provide biological context; drug–target interactions (from databases such as STITCH, BindingDB, and ChEMBL); and disease-related genes/proteins from and disease-related genes/proteins (from GWAS and transcriptomic studies). Topological analyses—using metrics such as degree and betweenness centrality—allow for the identification of critical hub nodes and functional modules, which are often ideal targets for multi-target drug design.

While this static network approach represents a monumental leap from single-target thinking and remains a foundational tool, it ultimately provides only a static snapshot of a dynamic system. It inherently overlooks the temporal evolution of disease networks across different stages and their specific manifestations within distinct biological contexts, such as cell types or brain regions. Addressing these limitations is the primary goal of next-generation network pharmacology, which leverages artificial intelligence to model the full complexity of living systems [34,35].

#### 3.2.2. Next-Generation Network Pharmacology: From Static Target Prediction to Dynamic, Context-Aware Network Construction

Building upon foundational static models, next-generation NP moves beyond the snapshot view by explicitly incorporating temporal dynamics and biological context into network construction. This shift is powered by the integration of artificial intelligence, which provides the computational tools necessary to infer and model these complex relationships. Advances are primarily unfolding in three interconnected areas:

##### Dynamic Network Modeling: Capturing Temporal Causality

Static networks reveal association, but dynamic models aim to infer causality and temporal progression. This is achieved by integrating temporal ML models with longitudinal multi-omics data (e.g., repeatedly sampled microbiomes, metabolomes, or transcriptomes over a treatment course) [36]. These models can simulate how a perturbation (e.g., a drug, a stressor) propagates through the molecular network over time, revealing cascading effects and critical intervention points. A compelling example is provided by Scutari et al., who employed dynamic Bayesian Networks (DBNs) to investigate the bidirectional causal relationships between skin diseases and mental disorders at an infodemiological level. Their model, which accounted for spatiotemporal dependencies and utilized bootstrap resampling for robust inference, successfully identified key skin-to-brain and brain-to-skin interaction pathways, demonstrating the unique capability of DBNs to capture reciprocal interactions across time points—a fundamental limitation of static network models [37]. Nevertheless, fundamental challenges in computational psychiatry—notably, the standardization and reproducibility of complex analytical workflows—persist.

##### Context-Specific Integration: Modeling Spatial and Cellular Heterogeneity

Biological networks are not universal; they vary widely across cell types, tissue microenvironments, and brain regions. Graph Neural Networks (GNNs) provide a powerful computational framework for this context-aware integration, enabling the merger of the core “drug–target–disease” network with multilayered contextual information. This includes leveraging single-cell RNA-seq data to infer cell-type-specific interaction networks, integrating spatial transcriptomics and proteomics data to decode the influence of cellular spatial organization on network topology, and incorporating compound properties to predict polypharmacy effects within a specific pathological context [38,39,40]. This approach aims to construct networks tailored to particular pathological states rather than employing a “one-size-fits-all” model, thereby paving the way for true precision psychiatry.

However, the effectiveness of GNNs in this domain is fundamentally constrained by a critical prerequisite: the quality and representativeness of the input network. A core and often underappreciated challenge is graph construction bias [41]. The performance and interpretability of GNNs are profoundly sensitive to how nodes and edges are defined in the initial biological network. This process is itself a hypothesis-driven modeling choice based on incomplete prior knowledge (e.g., protein–protein interaction databases), not an objective ground truth. Consequently, a biased or incomplete starting graph directly leads to biased model predictions and interpretations. This fundamental limitation, along with the broader methodological challenges it precipitates, is systematically elaborated in the section “Challenges and Future Directions for Next-Generation NP”.

##### Multi-Layered Network Fusion: Constructing Supernetworks of Systemic Dysregulation

The most holistic approach involves constructing multi-layered super-networks (or heterogeneous graphs) that vertically integrate disparate biological scales—genomic variation, epigenetic marks, gene expression, protein activities, and metabolic phenotypes—into a unified graph [42]. AI, particularly GNNs, is tasked with parsing the nonlinear relationships among these layers to identify cross-omic functional modules and key intermediary nodes that drive systemic dysregulation in MDD. This moves far beyond conventional PPI networks.

The construction of such multi-layered supernetworks represents a powerful approach to modeling MDD’s systemic dysregulation. However, the biological plausibility and utility of these AI-inferred networks face fundamental methodological challenges, particularly the sensitivity of their structure to initial assumptions and the lack of a definitive validation benchmark. To address this, we advocate for a multifaceted validation strategy (detailed in the section “Challenges and Future Directions for Next-Generation NP”) that combines in silico techniques, such as perturbation analysis, with ultimate experimental validation through wet-lab methods to establish causal biological plausibility.

##### Challenges and Future Directions for Next-Generation NP

Despite its promise, implementing next-generation network pharmacology faces several fundamental challenges that must be acknowledged.

First, data scarcity and heterogeneity pose a foundational bottleneck. Constructing dynamic and context-specific models demands vast amounts of high-quality, longitudinal, multi-modal data, which are often scarce, noisy, and costly to generate [43].

Second, graph construction bias remains a core methodological constraint. As discussed in the section “Context-Specific Integration: Modeling Spatial and Cellular Heterogeneity”, the structure and predictions of any network model are profoundly limited by the completeness and accuracy of the prior knowledge (e.g., from interaction databases) used to define the initial graph [44]. Models trained on biased or incomplete “prior graphs” risk producing misleading outputs.

Third, the validation of inferred networks lacks a “gold standard”. There is no absolute ground truth against which to benchmark computationally derived interactions [45]. Consequently, adopting a multifaceted validation strategy is imperative. This integrated approach comprises two critical components: Firstly, in silico validation employs techniques such as network perturbation analysis to assess the concordance between model-predicted key nodes and well-established essential genes or drug targets from independent databases [46]. Secondly, experimental validation is essential, wherein hypotheses concerning critical hubs and pathways require functional confirmation through wet-lab experiments, typically involving CRISPR-based knockdown or knockout in relevant cell models to establish causal roles [47].

Finally, model interpretability and generalizability pose significant translational barriers. The “black box” nature of complex models such as GNNs impedes clear biological interpretation [48]. More critically, models trained on specific cohorts often exhibit poor cross-cohort generalizability due to technical batch effects and population heterogeneity, limiting their broad clinical applicability.

### 3.3. AI as an Integrative Engine: From Multi-Omics Data to Mechanistic Insights in Depression

Understanding major depressive disorder as a complex systems-level disorder necessitates the integration of high-dimensional, multi-omics data. This integration poses significant analytical challenges related to dimensionality, heterogeneity, and dynamic complexity. AI serves as the essential engine for this integration, transforming these challenges into opportunities to generate novel, testable hypotheses about disease mechanisms. Rather than just a predictive toolkit, AI provides a framework to decipher the biological heterogeneity underlying clinical symptoms, model the dysregulated networks central to pathophysiology, and semantically bridge disparate data modalities—collectively advancing the field toward precision psychiatry.

#### 3.3.1. Machine and Deep Learning: From Data Integration to Mechanistic Hypotheses

AI, particularly ML and DL, provides the essential computational framework for integrating high-dimensional, heterogeneous multi-omics data in depression research. Its value lies not merely in prediction, but in its capacity to generate data-driven, mechanistic hypotheses about disease heterogeneity.

Unsupervised learning addresses the core challenge of disease subtyping beyond symptomatic diagnosis. By identifying data-driven biotypes from integrated omics layers, these methods directly probe the biological heterogeneity underlying the clinical syndrome of MDD. For instance, hierarchical clustering of neurophysiological data by Drysdale et al. identified subtypes with distinct treatment response profiles, suggesting divergent underlying circuit dysfunctions [49]. Such approaches move the field towards a biology-based nosology.

Supervised learning addresses the challenge of integrating diverse biomarkers to improve prediction. By fusing features across genomic, epigenomic, and other domains, these models can improve the prediction of diagnosis or treatment outcome [50,51]. Critically, the feature importance metrics from models like Random Forests can highlight specific biological pathways (e.g., inflammation-related genes or specific methylation sites) worthy of further mechanistic investigation, thus bridging prediction and biological insight.

DL and GNNs offer a powerful paradigm for modeling complex, relational biological systems. In depression research, GNNs can analyze brain functional connectomes or molecular interaction networks to identify aberrant connectivity patterns or key pathogenic hubs, shifting the focus from individual biomarkers to dysfunctional systems-level properties [52]. Recent methodological innovations underscore this potential; for example, the Graph Frequency Attention Convolutional Neural Network incorporates electroencephalogram (EEG) signal frequency as an attention mechanism to prioritize clinically relevant neural rhythms. This model has shown superior performance in classifying treatment responses compared to established benchmarks, highlighting the utility of advanced GNNs in decoding complex brain dynamics and supporting clinical decisions [52].

#### 3.3.2. Natural Language Processing: Bridging Subjective Experience and Biological Constructs

Natural Language Processing (NLP) has emerged as a pivotal tool for extracting quantifiable insights from unstructured textual data, offering a unique bridge between subjective patient experiences and objective biological measures in depression research [53]. By analyzing diverse digital footprints—from clinical notes and therapy transcripts to social media posts—NLP can identify subtle linguistic markers (e.g., emotional valence, semantic coherence, and first-person pronoun usage) that serve as digital phenotypes of depressive states [54,55].

While conventional machine learning models (e.g., Support Vector Machines, decision trees) provided foundational classification capabilities [54], the advent of large language models (LLMs) has markedly advanced the field. LLMs capture deeper contextual and semantic features specific to mental health language, thereby significantly improving the accuracy and generalizability of depression detection and progression-monitoring systems [55].

Critically, the value of NLP extends beyond standalone digital phenotyping. Its power is fully realized when linguistic features are integrated with other biological modalities. This multi-modal integration is exemplified by Dougherty et al., who combined NLP-derived features with electroencephalographic imaging to predict longitudinal responses to psilocybin therapy in treatment-resistant depression. Their model achieved high cross-validated accuracy, demonstrating that the combination of language-based and neurophysiological data outperforms either modality alone [56]. This approach underscores a central paradigm: NLP transforms qualitative narrative into structured data, enabling its correlation with genomic, metabolomic, or neuroimaging findings to construct a more comprehensive, multi-dimensional model of depression.

Thus, NLP operates not merely as an analytical tool but as a critical semantic translator, weaving a patient-generated narrative into the broader tapestry of multi-omics data. This integration is essential for advancing a precision psychiatry framework that is both biologically grounded and intimately informed by the lived experience of illness.

#### 3.3.3. Generative AI: As a Novel Engine for Hypothesis Generation and Data Augmentation

Beyond its clinical applications, Generative AI (GAI) offers a transformative potential for mechanistic research in depression by addressing two fundamental bottlenecks in multi-omics integration: the scarcity of large-scale, well-annotated datasets and the difficulty in generating causal, testable hypotheses from high-dimensional data. Techniques such as generative adversarial networks and variational autoencoders can create high-fidelity synthetic multi-omics data [57], thereby augmenting limited clinical cohorts and enhancing the robustness of downstream analytical models [58]. This capability is critical for powering data-intensive deep learning approaches without exacerbating concerns of overfitting or compromising patient privacy.

More profoundly, GAI serves as a powerful engine for knowledge synthesis and the generation of mechanistic hypotheses. Researchers can leverage large language models (LLMs), pre-trained on vast biomedical corpora, to mine the literature and infer novel connections between disparate omics data, clinical symptoms, and known pathophysiology [59]. For instance, a GAI system could be prompted to propose novel gene regulatory networks underlying treatment-resistant depression by integrating transcriptomic data with known protein–protein interactions and pharmacogenetic databases, thus yielding specific, experimentally verifiable hypotheses.

This approach is exemplified by emerging studies employing GAI to design novel multi-epitope vaccines for major depression based on integrated genomic and proteomic data [60], or to predict the functional impact of non-coding genetic variants on neuronal gene regulation [61]. These applications demonstrate how GAI moves beyond pattern recognition to actively propose biomolecular mechanisms and therapeutic candidates that might remain obscured through conventional analysis.

Nevertheless, integrating GAI into the multi-omics research pipeline poses significant domain-specific challenges. These include the risk of generating biologically plausible but factually incorrect hallucinations [62], the embedded biases of training data that may perpetuate existing biomedical disparities [63], and the critical need for rigorous in vitro and in vivo validation of any AI-generated hypothesis. Future work must therefore prioritize the development of transparent, explainable, and biologically constrained GAI frameworks that can be seamlessly integrated into the iterative cycle of computational prediction and experimental validation, ultimately accelerating the path toward novel therapeutic targets.

A consolidated overview of these core AI approaches, their respective integrative functions, and exemplary applications is presented in Table 2.

### 3.4. Data Integration and Analytical Framework: From Data to Systems-Level Insights

The central challenge in multi-omics research lies in integrating heterogeneous data types—spanning genomics, transcriptomics, epigenomics, and proteomics—with structured biomedical knowledge and artificial intelligence algorithms into a unified analytical framework. Among contemporary computational approaches, GNNs have emerged as one of the most promising avenues for this integration [64]. Their capability to natively represent biological systems as graphs—where nodes denote biological entities (e.g., genes, proteins, or metabolites) and edges represent functional or physical interactions—makes them particularly suitable for modeling complex, relational biomedical data.

#### 3.4.1. Graph Neural Networks: A Technical Foundation

GNNs constitute a class of deep learning models designed to operate directly on graph-structured data. Architectures such as Graph Convolutional Networks, GraphSAGE, and Graph Attention Networks have achieved state-of-the-art performance across numerous bioinformatics tasks, including disease subtype prediction, drug repurposing, and biomarker discovery [65]. Initially developed for node-level classification, GNNs leverage message-passing mechanisms to propagate and aggregate feature information across adjacent nodes, enabling effective prediction and representation learning even with highly sparse and interconnected biological data [66].

#### 3.4.2. Information Fusion and Inference via GNNs

Methodological advances have extended GNNs to multi-relational and heterogeneous graphs, facilitating the integration of diverse omics data types—such as transcriptomic, epigenomic, and spatial transcriptomic profiles—within a single model. Incorporation of external biological networks (e.g., protein–protein interactions, gene regulatory networks, and ligand–receptor databases) further enhances the contextual relevance and interpretability of GNN predictions. These integrative approaches allow models to infer functional biological relationships—such as regulatory mechanisms or intercellular signaling pathways—that are not readily apparent from omics data alone [65]. Nevertheless, several computational and practical challenges impede the broad application of GNNs in real-world biomedical contexts.

#### 3.4.3. Current Challenges and Computational Strategies

A key limitation of many GNN architectures is their scalability to large biological graphs, constrained by high computational and memory demands. Furthermore, many models exhibit excessive reliance on local neighborhood information, constraining their ability to capture global graph topology and long-range dependencies—a critical shortcoming in applications such as genome-scale regulatory network inference. To mitigate this, one common strategy is to design deeper network architectures or incorporate hierarchical pooling mechanisms, thereby expanding the receptive field and promoting the integration of information from broader network contexts [65].

#### 3.4.4. A Practical Roadmap for GNNs in Depression Research

Translating the theoretical strengths of GNNs into practical tools for depression research requires a domain-specific methodological framework that addresses salient challenges such as data heterogeneity, temporal sparsity, and interpretability.

The suitability of GNNs for multi-omics integration in neuropsychiatric research stems from their ability to explicitly incorporate biological priors—such as protein interaction networks or neuroimaging-derived connectomes—into an inductive learning framework [67]. This facilitates biologically grounded generalization, which is especially valuable in settings with limited sample sizes. Empirical validation of GNN-based approaches should include systematic benchmarking against alternative deep learning models, including transformers and multimodal autoencoders, using metrics such as predictive accuracy, robustness to noise, and interpretability [68,69]. Furthermore, the development and evaluation of hybrid or co-learning models that integrate the strengths of both paradigms represent a promising avenue for next-generation integrative analysis [70].

Constructing robust graph representations for MDD necessitates a comprehensive pre-processing and integration strategy to address data heterogeneity. This includes independent batch-effect correction for each omics layer prior to graph construction [71], incorporating adversarial domain adaptation within the GNN to minimize technical variability, and handling missing data via graph-aware imputation techniques [72] or multi-task architectures that jointly learn to impute missing values and predict clinical outcomes. To improve generalizability across independent cohorts or sequencing platforms, frameworks such as graph meta-learning or invariant representation learning can be employed to derive domain-agnostic features [73].

Model interpretability remains paramount when applying GNNs to high-dimensional, noisy biological data. Beyond using intrinsically interpretable architectures like GATs—which provide attention-based importance scores for nodes and edges—post hoc interpretation tools (e.g., GNNExplainer) can identify minimal predictive subgraphs that can be functionally annotated via pathway enrichment analysis [74]. Incorporating sparsity-inducing constraints during training can also promote simpler and more interpretable model structures.

Finally, modeling temporal dynamics in longitudinal multi-omics data—often sparse in MDD studies—requires specialized approaches. Techniques such as neural ordinary differential equations or Gaussian processes can reconstruct continuous individual trajectories from sparse observations, enabling the construction of dynamic graphs. Alternatively, temporal GNNs or dynamic embedding methods can directly model irregularly sampled time-series omics data. Incorporating ensemble learning strategies may further enhance the robustness and generalizability of dynamic forecasting models [75]. Together, these strategies form a comprehensive analytical workflow for translating multi-omics data into actionable insights within depression research.

## 4. A Systems-Level Case Study: The Gut–Brain Axis in Depression Pathophysiology

### 4.1. Identification of Key Molecular Modules and Pathways

Integrated multi-omics analyses have revealed core signaling pathway networks that extend beyond the classic monoaminergic hypothesis [76]. These include, but are not limited to, neuroinflammation [10], mitochondrial dysfunction [77], circadian rhythm disruptions [78], and neurotrophic factor signaling [79]. Collectively, these findings underscore the complex, interconnected biological architecture underlying depression and provide new mechanistic entry points for therapeutic intervention.

### 4.2. Parsing Disease Subtypes (Biotypes)

Unsupervised ML approaches applied to multi-omics data have enabled the identification of depression subtypes—referred to as biotypes—characterized by distinct molecular signatures [80]. These data-driven classifications move beyond symptom-based diagnoses and offer a biological foundation for precision subtyping, potentially guiding more targeted and effective treatment strategies.

### 4.3. The Gut–Brain Axis: A Central Interface

The gut microbiota is a highly complex, dynamically evolving consortium of diverse microorganisms—including bacteria, fungi, parasites, and viruses—that exhibits considerable geographic variation. These microbial communities engage in extensive intra- and interspecies interactions, as well as symbiotic relationships with the host, and play an essential role in maintaining normal physiological homeostasis [81]. The gut microbiota significantly modulates cognitive processes and behaviors via the MGB axis, an integrated network connecting the gastrointestinal tract and the central nervous system via neural, immune, and endocrine signaling mechanisms [82,83].

#### 4.3.1. Neuroimmune and Neuroinflammatory Pathways

Building on the MGB axis, dysbiosis-induced systemic inflammation is a key mechanistic pathway linking the MGB axis to depression. An imbalanced gut microbiota can trigger the release of pro-inflammatory cytokines, which may cross the compromised blood–brain barrier and contribute directly to the development of depression symptoms [84]. A compelling mechanistic link involves matrix metalloproteinase-8 (MMP8), which is elevated in MDD patients and stress-vulnerable mice. MMP8 can traverse the blood–brain barrier, localize to reward-related regions such as the nucleus accumbens, and remodel the extracellular matrix, ultimately impairing social reward behavior [85,86]. This state of chronic inflammation is increasingly recognized as a central factor in depression pathophysiology, capable of damaging neuronal tissue and disrupting function [87].

#### 4.3.2. Neurotransmitter and Metabolic Regulation

Beyond neuroimmune mechanisms, gut microbiota dysbiosis significantly influences neurotransmitter systems critically involved in MDD pathophysiology. A substantial proportion of neurotransmitters—including serotonin (5-HT), norepinephrine, dopamine, γ-aminobutyric acid (GABA), glutamate, and acetylcholine—or their precursors are produced by gut microbes or gut-enteroendocrine cells. These molecules enter the systemic circulation and subsequently affect brain function either by crossing the blood–brain barrier or by activating vagus nerve-mediated signaling [88,89].

Microbiota-derived metabolites further contribute to MDD through several parallel pathways. Short-chain fatty acids (SCFAs) modulate depressive-like behaviors via: (1) regulating the HPA axis [90]; (2) enhancing central serotonin synthesis via tryptophan metabolism [91]; and (3) exerting anti-inflammatory effects and reinforcing intestinal barrier integrity [92]. Additional microbial metabolic pathways implicated in MDD include bile acid metabolism and dysregulation of the tryptophan-kynurenine (Trp-Kyn) pathway, both of which may influence neuroactive signaling and immune activation [93,94].

#### 4.3.3. Neuroplasticity and Structural Integration

In addition to systemic inflammation and metabolic regulation, the MGB axis influences depression pathophysiology through direct effects on neuroplasticity. Gut-derived signals, potentially mediated by microbial metabolites such as SCFAs, can regulate microglial development and function, thereby shaping neural circuitry and behavior [95]. Furthermore, impaired adult hippocampal neurogenesis—evidenced by reduced neural stem/progenitor cells, decreased granule neuron numbers, and lower dentate gyrus volume—represents a consistent morphological correlate of MDD [96,97,98]. The gut microbiota can modulate adult hippocampal neurogenesis and stress-related hormone secretion via the HPA axis, thereby participating in the regulation of mood and cognition [99]. These interrelated pathways are summarized in Figure 2.

Compelling evidence for the microbiota’s essential role comes from germ-free animal models; for instance, germ-free mice exhibit an immature enteric nervous system and defective sensory signaling, deficits which can be reversed by restoring a normal gut microbiota [100,101]. Clinically, a substantial proportion of MDD patients present with somatic manifestations indicative of MGB axis dysregulation, commonly encompassing gastrointestinal complaints, appetite loss, nausea, and metabolic irregularities [102]. Ultimately, integrating these mechanisms into molecular network models provides a more comprehensive understanding of depression’s systemic nature.

#### 4.3.4. Evidence Heterogeneity and Translational Challenges

While preclinical evidence robustly supports the involvement of the MGB axis in depression pathophysiology, translating this evidence into effective clinical interventions remains complex and poses significant methodological challenges. Notably, several high-profile randomized controlled trials of probiotic supplementation (psychobiotics) have failed to demonstrate substantial antidepressant effects over the placebo in broad MDD populations [103,104]. This discrepancy underscores the complexity of translating mechanistic knowledge into effective therapies. Potential explanations for these mixed results include the vast heterogeneity in probiotic strains, formulations, and doses used across studies, which may not optimally target specific dysbiotic states or pathways relevant to depression. Furthermore, considerable inter-individual variability in baseline gut microbiota composition and host physiology suggests that effective interventions may need to be personalized rather than universal [105]. Adding to this complexity is the current lack of standardized treatment protocols. This is particularly relevant for patients with severe or chronic MDD, for whom modulating the gut microbiota alone may be insufficient, potentially necessitating combination with other therapeutic modalities [106].

These outcomes do not invalidate the MGB hypothesis but rather emphasize that modulating a complex, individualized ecosystem requires precision. Future research must therefore focus on biomarker-guided patient stratification (e.g., identifying inflammation- or tryptophan metabolism-dominant biotypes) and on testing targeted, next-generation psychobiotics or microbiome-based therapies in mechanistically aligned subpopulations. The integration of multi-omics profiling into clinical trial design is essential to identify responders and refine therapeutic strategies.

## 5. Translation and Application: From Big Data to Novel Diagnostic and Therapeutic Strategies

The rapidly evolving understanding of mechanisms of the MGB axis is fostering a transformative paradigm in translational psychiatry, enabling the conversion of multidimensional data into innovative diagnostic and therapeutic solutions. This paradigm shift is powered by the synergistic integration of multi-omics technologies with artificial intelligence, creating new pathways for biomarker discovery and precision medicine in depression. The computational foundation for this transformation is established through a suite of AI methodologies—from machine learning algorithms for pattern extraction to graph-based architectures for modeling biological networks—that collectively enable the systematic analysis of complex disease mechanisms and support the development of targeted interventions.

### 5.1. Biomarker Discovery and Validation

A single biomarker is insufficient to capture the high heterogeneity of MDD. Consequently, research strategies are shifting towards integrating multidimensional information from genomics, metagenomics, metabolomics, proteomics, and clinical data to discover and validate composite biomarker panels.

The integration of diverse high-dimensional datasets offers a powerful strategy for deriving multi-omics biosignatures, which can elucidate the biological underpinnings of clinical heterogeneity in depression. Specifically, Data Integration Analysis for Biomarker discovery using Latent components (DIABLO) employs a modified sparse partial least squares discriminant analysis to build predictive multi-omics signatures. These signatures are derived from the correlations within and between datasets to classify group membership, and the method applies to large, diverse datasets regardless of prior feature selection [17]. The identification of such data-driven biosignatures not only refines diagnostic precision but also directly informs the development of targeted therapeutics and personalized intervention strategies, as discussed in the following sections.

### 5.2. Drug Discovery and Repurposing

The discovery of the MGB axis has opened novel avenues for antidepressant drug development, with a core philosophy shifting from traditional “single-target” approaches to “network regulation” that better reflects the disease’s complexity.

#### 5.2.1. Novel Multi-Target Drug Design

Building on the comprehensive pathophysiological framework of the MGB axis established in Section 3.3, NP analysis enables the identification of key, dysregulated nodes within this complex system. Novel therapeutic strategies aim to modulate multiple key nodes within this dysregulated network concurrently. The following approaches exemplify this rational design principle:

Targeting Neuroimmune Crosstalk: The established role of peripheral mediators, such as MMP8, presents a compelling therapeutic target. Rational drug design could aim to develop selective MMP8 inhibitors to prevent its detrimental remodeling of the nucleus accumbens extracellular matrix, thereby preserving social reward behavior.

Modulating Microbial Neurotransmitter Synthesis: The gut’s significant contribution to peripheral neurotransmitter precursors offers a novel avenue for intervention. One strategy involves developing small-molecule modulators that selectively influence microbial enzymes involved in the synthesis of tryptophan, tyrosine, or GABA.

Harnessing Metabolite Signaling Pathways: The beneficial effects of SCFAs via receptor-mediated mechanisms provide a blueprint for developing receptor agonists. Designing stable, bioavailable FFAR2/3 agonists could replicate the anti-inflammatory, barrier-strengthening, and HPA-axis-modulating effects of SCFAs. Similarly, correcting Trp-Kyn pathway dysregulation through Indoleamine 2,3-dioxygenase/Tryptophan 2,3-dioxygenase inhibitors represents a promising multi-target strategy.

Promoting Neuroplasticity: Future therapies may integrate the aforementioned strategies with agents that directly promote neurotrophic activity and neuronal resilience, thereby targeting the core morphological deficits of the disorder.

Collectively, these strategies establish a conceptual framework for multi-target intervention. Its practical realization critically depends on integrating network pharmacology with computational modeling to systematically decipher the interaction networks and pinpoint the optimal multi-target ligand combinations or repurposing opportunities for experimental testing.

#### 5.2.2. AI-Driven Drug Repurposing

Drug repurposing is a cost- and time-efficient strategy that leverages existing knowledge of drug safety, pharmacokinetics, and manufacturing processes to identify new therapeutic applications, offering a viable alternative to traditional de novo drug development [107]. The integration of AI has transformed this strategy from a serendipitous process into a systematic, data-driven paradigm [108]. AI-powered computational frameworks can integrate multi-source biomedical data to efficiently predict novel drug-disease associations. This advancement is exemplified by high-throughput systems such as TensorFlow Drug Repurposer and DeepDrugRepurposing, which enable large-scale predictions by systematically analyzing multidimensional drug screening datasets [109].

The efficacy of these AI platforms stems from a suite of sophisticated computational and experimental methodologies. Computational methodologies include feature matching based on gene expression profiles (transcriptomics), metabolomics and proteomics, chemical structure similarity analysis, adverse drug reaction profiling, and molecular docking and GWAS. Complementary experimental approaches include retrospective clinical data analysis (e.g., electronic health records and post-marketing surveillance), the use of novel data sources (such as cancer cell line screening and patient-reported outcomes), target-binding assays, and phenotypic screening. Through these integrated strategies, researchers can systematically identify novel therapeutic indications for existing drugs in the treatment of depression across multiple dimensions [110].

A key advantage of the AI-driven approach is its ability to significantly accelerate the identification of repurposing candidates and streamline their progression into Phase II/III trials. By leveraging pre-existing pharmacological and toxicological data, AI models can accurately predict a drug’s efficacy and safety profile in new disease contexts, thereby substantially reducing both development timelines and associated costs [111].

In the context of depression, AI-driven repurposing has broadened the therapeutic landscape, with numerous agents across diverse therapeutic classes currently under investigation or approved based on computational predictions [110,112]. For instance, AI-based analyses of electronic health records and molecular interaction networks have suggested repurposing potential for various drug categories in bipolar disorder and depression. Successful examples include at least thirteen non-psychiatric drugs and twenty antidepressants that have been assigned new indications through computationally guided approaches [112].

AI models demonstrate particular efficacy in evaluating and prioritizing repurposing candidates from broad pharmacological categories, including anesthetics, GABA receptor modulators, second-generation antipsychotics, dopamine agonists, and N-Methyl-D-aspartic acid (NMDA) receptor antagonists. Furthermore, AI has identified potential antidepressant activity in seemingly unrelated drug classes, such as antibiotics, antidiabetics, Hydroxy Methyl Glutaryl-CoA reductase inhibitors (statins), angiotensin-converting enzyme inhibitors, and calcium channel blockers. These diverse agents, predicted to act via multi-target mechanisms, are hypothesized to exert antidepressant effects through pathways that align with their known pharmacological classes—a connection revealed through AI-driven pattern recognition and pathway analysis [113].

#### 5.2.3. Modernization of Natural Products and Herbal Formulations

Natural products mediate their antidepressant effects through the synergistic actions of multiple bioactive constituents, which collectively target diverse molecular pathways to regulate the MGB axis in MDD [114].

Flavonoids, a class of polyphenolic compounds widely distributed in plants, exhibit a broad spectrum of biological activities, including antioxidant, anti-inflammatory, and neuroprotective properties [115]. Preclinical studies indicate that specific flavonoids exert antidepressant-like effects in animal models of depression [116]. For example, quercetin modulates key neurotransmitter systems such as NMDA and GABA receptors, downregulates pro-inflammatory cytokines (TNF-α, IL-1β), and activates the Nrf-2/ARE antioxidant pathway. It also influences brain-derived neurotrophic factor (BDNF) signaling and intracellular cascades, including PI3K/Akt and MAPK/ERK, underscoring its multi-target neuroprotective potential [117]. Puerarin, a major flavonoid derived from the root of *Pueraria montana* var. *lobata* (Ohwi) Maesen and S. M. Almeida, demonstrates neuroprotective efficacy by improving microcirculation and counteracting inflammatory and oxidative stress [118]. In Chronic Unpredictable Mild Stress (CUMS)-induced depressive rats, puerarin restored gut microbiota composition, reduced pathogenic bacteria such as *Desulfovibrio* and *Verrucomicrobia*, and enhanced hippocampal BDNF and IκB-α expression. It suppressed NF-κB activation, collectively contributing to its antidepressant effect [119].

Beyond flavonoids, alkaloids constitute another major class of plant-derived compounds that exhibit multi-target antidepressant effects via the MGB axis [120]. For instance, berberine, the primary bioactive alkaloid from *Coptis chinensis* Franch, modulates gut microbial composition by elevating Firmicutes and reducing Bacteroides, thereby altering SCFA profiles—specifically suppressing acetic and propionic acids while increasing isovaleric acid. These changes correlate with activation of the MGB axis, elevated hippocampal BDNF and monoamine levels, and alleviation of depressive-like behaviors [121].

Natural polysaccharides have attracted increasing interest due to their favorable safety profile and multi-target mechanisms, including antidepressant effects mediated through gut–brain communication [122]. *Schisandra chinensis* polysaccharide ameliorates depressive behaviors by inhibiting HPA axis overactivation and hepatic oxidative stress, restoring neurotrophic and synaptic plasticity, and modulating gut microbiota [123]. Similarly, *Corydalis yanhusuo* polysaccharides enhance monoamine neurotransmitters (Norepinephrine, Dopamine, 5-HT) and BDNF expression by regulating gut microbiota structure and SCFA metabolism while upregulating TPH-2 to mitigate neuronal damage in CUMS models [124].

Cryptotanshinone, a quinoid diterpene from *Salvia miltiorrhiza* Bunge, alleviates depression-like phenotypes by attenuating systemic inflammation and rebalancing gut microbiota (e.g., reducing *Parabacteroides merdae*), partly through modulation of the PI3K-AKT pathway [125]. Apple polyphenol extract rich in epicatechin and chlorogenic acid restores gut microbial homeostasis, inhibits NF-κB activation, enhances expression of tight junction proteins (occludin, ZO-1) in the gut and hippocampus, and normalizes HPA axis dysregulation, collectively contributing to its antidepressant efficacy [126]. Polygalae radix oligosaccharide esters increase beneficial gut bacteria (e.g., Actinobacteria and Firmicutes), reinforce intestinal barrier integrity, and modulate the tryptophan-kynurenine pathway to regulate brain neurotransmission [127]. *Cuscutae Semen* extract, whose primary active constituents include chlorogenic acid and hypericin, exhibits considerable antidepressant and anti-inflammatory activities. Studies indicate that these compounds ameliorate depressive-like behaviors induced by chronic unpredictable stress in mice, potentially via modulation of the gut microbiota, inhibition of NLRP3/NF-κB-mediated neuroinflammatory pathways, and preservation of synaptic integrity [128]. Saffron extract, derived from *Crocus sativus* L., increases the abundance of beneficial gut microbiota such as *Akkermansia* and *Muribaculaceae flora*, which are inversely associated with the neurotoxic metabolite dimethylamine. By reducing dimethylamine levels and modulating brain proteomic profiles, saffron extract exerts notable antidepressant effects [129]. The multi-target mechanisms of representative natural compounds targeting the MGB axis for antidepressant effects are systematically summarized in Table 3.

In summary, these naturally derived compounds target the MGB axis through microbiota remodeling, anti-inflammatory and antioxidant mechanisms, and neuroendocrine regulation, offering promising scaffolds for the development of novel antidepressant agents. This approach provides a modern scientific interpretation for traditional medical knowledge and offers valuable molecular blueprints for new drug development.

### 5.3. Towards Precision Psychiatry

The long-term goal of precision medicine in depression is to transcend the current “one-size-fits-all” treatment paradigm. A promising, albeit aspirational, pathway lies in leveraging multi-omics data to create dynamic, in silico models of an individual’s pathophysiological state. These models, which aim to emulate the patient’s unique biological function, can be continuously updated with new data to simulate disease progression and virtually test interventions. This concept, sometimes described as building a “clinical digital twin” or a “digital self” [130], represents the ultimate endpoint of this endeavor.

Despite these challenges, a foundational framework is emerging. By integrating a patient’s gut metagenomic, plasma metabolomic, immunomic, and genomic data within probabilistic or mechanistic models, it becomes possible to generate hypotheses about underlying biological subsystems and their perturbation [131]. For instance, machine learning models trained on multi-omics datasets have already demonstrated the ability to stratify patients into biotypes with distinct inflammatory or metabolic profiles, which are predictive of treatment outcomes [49,132]. These models can be used in silico to simulate the potential effects, functioning not as definitive predictions but as data-driven decision-support tools.

The actionable output of such a system is a ranked set of personalized, evidence-based hypotheses [133]. For example, for a patient subgroup identified with significant gut dysbiosis and a plasma metabolome indicative of low-grade inflammation, the model might prioritize testing adjunctive anti-inflammatory nutraceuticals or specific probiotic regimens, as biomarker-defined inflammation has been shown to predict response to anti-inflammatory therapy in depression [105]. For those with profiles suggesting severe deficiencies in microbial metabolite production (e.g., SCFAs), a high-fiber dietary intervention might be a primary focus, given the established role of prebiotics in modulating gut–brain axis pathways relevant to depression [134]. This framework does not propose replacing clinician judgment but to augment it with insights derived from complex, high-dimensional data [135]. Its iterative refinement and validation through rigorous prospective clinical studies are the essential next steps for translation [135]. To delineate this translational pathway, Table 4 summarizes the principal translational applications of AI in depression management, encompassing core tasks, data types, and representative computational methods, thereby providing researchers with a systematic reference framework.

## 6. Current Challenges and Future Perspectives

### 6.1. Challenges at the Data Level

The development and effectiveness of AI models are critically dependent on high-quality, well-structured, and comprehensive datasets. However, acquiring such data remains a formidable challenge. Existing datasets commonly suffer from limitations in accessibility, poor data quality, and internal inconsistencies, all of which compromise model reliability and lead to biased predictions [136]. The issues of data scarcity and heterogeneity are particularly pronounced in chemical medicine, where they significantly constrain the accuracy and generalizability of predictive models [137,138]. Techniques such as data augmentation and synthetic dataset generation offer viable pathways to address this bottleneck, effectively expanding dataset scale and diversity, thereby enhancing model performance. Nevertheless, these technical approaches must be applied cautiously to avoid introducing new biases that could further undermine model integrity [139].

### 6.2. Technical Challenges

A paramount technical challenge in AI involves the inherent opacity and limited interpretability of complex models, particularly DL architectures. These systems are frequently characterized as “black boxes” due to the considerable difficulty in elucidating their internal decision-making processes [140]. Despite their capacity to generate highly accurate predictions, the underlying rationale behind these outputs often remains obscure, raising significant concerns about the reliability and fairness of AI-driven predictions and decisions [111]. This lack of transparency poses a substantial barrier to adoption in critical sectors such as healthcare and finance, where understanding the basis for a decision is essential.

Furthermore, for network pharmacology, transitioning from relatively static “drug–target–disease” networks to dynamic, computable disease models constructed through the integration of AI and multi-omics data calls for the development of new, more sophisticated network analysis and validation tools.

To establish trust, ensure accountability, and facilitate regulatory compliance, AI models must exhibit a degree of interpretability accessible to stakeholders, including end-users and oversight bodies. The ability to audit and comprehend computational decision-making processes is a fundamental prerequisite for verifying fairness, identifying potential biases, and assuring reliability. While ongoing research continues to advance novel methodologies in XAI—such as the emerging Chemical-explainable Graph Neural Network, which demonstrates unique advantages in drug discovery [141]—achieving robust, universally applicable interpretability remains a significant, unresolved technical hurdle [111].

### 6.3. Biological Complexity

Multiple layers of biological complexity hinder the translation of biomedical discoveries into clinical applications. A central challenge lies in establishing causal relationships, which requires rigorously distinguishing correlation from causation [142]. This difficulty is compounded by the dynamic nature of biological processes, which are difficult to capture experimentally, and by the well-documented translational gap between animal models and human pathophysiology.

Although current AI systems excel at processing large-scale data, they remain limited in their ability to perform contextual reasoning and to adapt autonomously to novel or changing conditions—constraints that limit their effectiveness for modeling dynamic biological systems [111]. These limitations are starkly illustrated by the frequent failure of preclinical findings to translate to human outcomes. For example, Cataldi et al. report considerable discrepancies between animal and human studies of the gut–brain axis: mechanistic insights derived from highly controlled animal models often fail to generalize to heterogeneous human populations with diverse genetic backgrounds, diets, and lifestyles [143]. Thus, direct extrapolation from animal models to humans is inherently problematic. Future research should prioritize longitudinal human studies specifically designed to capture dynamic biological changes, rigorously control for confounders, and employ multi-omics technologies to bridge the translational gap and validate mechanistic pathways in humans [143].

### 6.4. Toward an Integrated Precision Psychiatry Pipeline

To translate the promise of AI and network pharmacology into clinical impact, future efforts must converge toward a coherent, patient-centered pipeline. This envisioned pipeline integrates causal discovery (through prospective multi-omics cohorts), dynamic monitoring (via digital phenotyping), and mechanism-targeted intervention (e.g., psychobiotics), all underpinned by interpretable AI and robust ethical frameworks.

#### 6.4.1. Foundational Cohort Studies for Causal Discovery

Future research must prioritize large-scale, prospective cohort studies that deeply integrate multi-omics profiling (genomics, metabolomics, proteomics) with detailed clinical and lifestyle data. These cohorts are essential for moving beyond correlations to establishing causal mechanisms. Methodologies such as Mendelian randomization should be employed to rigorously test causal hypotheses regarding modifiable factors (e.g., specific microbial taxa and dietary components) and depression risk [142,144]. Such studies will yield high-quality, temporally resolved datasets needed to train more robust, generalizable, and causality-aware AI models, ultimately identifying mechanistically defined disease biotypes and novel therapeutic targets.

#### 6.4.2. Dynamic Monitoring via Digital Phenotyping

Building on insights into emotional and cognitive variability preceding depressive episodes [145,146], the next step is to operationalize these findings for prevention. Future work should focus on developing multivariable dynamic models that synthesize data from wearables and smartphone sensors to track, in real-time, the central tendency, intra-individual variability, instability, and inertia of key states. The critical challenge is to translate these statistical patterns into clinically actionable, early warning systems. Success in this area will enable preemptive interventions for individuals at high risk, shifting the paradigm from reactive treatment to proactive management.

#### 6.4.3. Novel Microbiota-Targeted Intervention

The gut microbiota has emerged as a critical pathophysiological factor in depression. Restoration of healthy microbiota can be achieved through several strategies, notably via psychobiotic interventions—live probiotics and prebiotics that confer neurological benefits. Primary approaches include (1) probiotics, (2) prebiotics, (3) dietary interventions, and (4) fecal microbiota transplantation (FMT) [147].

Psychobiotics, which include beneficial probiotics and prebiotics, alleviate depressive and anxiety-like behaviors through multiple pathways. These encompass modulation of cognitive-emotional processes, regulation of the hypothalamic–pituitary–adrenal (HPA) axis, reduction of pro-inflammatory cytokines (e.g., IL-1β, TNF-α), and influence on key neurotransmitters and neurotrophic factors such as BDNF, GABA, and glutamate [148,149,150]. For example, administration of *Lactobacillus rhamnosus* JB-1 significantly reduced depressive and anxious behaviors in murine models [150,151]. Preclinical studies further demonstrate that specific probiotic strains can normalize HPA axis hyperactivity and enhance BDNF expression, thereby improving mood and cognitive function [152,153].

Prebiotics, predominantly derived from high-fiber foods, ameliorate stress-induced anxiety and modulate gut microbial composition and metabolism, which, in turn, influence central nervous system development and function [154]. Specific prebiotic compounds, including galacto-oligosaccharides and resistant starch, exhibit efficacy in animal models of anxiety and depression via gut–brain axis communication [155,156]. Additionally, cocoa-derived flavanols demonstrate antidepressant effects, likely due to their anti-inflammatory and antioxidant properties [157].

Dietary patterns significantly shape the gut microbiota and impact depressive symptoms. The Mediterranean diet—rich in olive oil, fruits, vegetables, and fermented dairy—attenuates oxidative stress and inflammation through microbiota-mediated mechanisms [158]. Very-low-carbohydrate ketogenic diets show potential in managing neurological disorders, including treatment-resistant depression [159]. Calorie restriction may also confer antidepressant effects via weight loss and reduced inflammation, though further clinical evidence is needed [160].

FMT restores ecological balance in the gut microbiome and holds therapeutic promise for depression, anxiety, and autism spectrum disorder.

As discussed above, multiple gut-targeted strategies show promise. A comparative overview of these interventions is presented in Table 5.

A combinatorial treatment strategy integrating psychobiotics, prebiotics, tailored diets, and FMT may constitute a novel and effective paradigm for managing depression [106].

#### 6.4.4. Ethical and Interpretable AI as Enabling Infrastructure

The entire pipeline depends on transparent and ethically governed AI tools. Advancing XAI methodologies is non-negotiable to ensure that model predictions (e.g., risk scores, treatment recommendations) are interpretable to clinicians and researchers, fostering trust and facilitating clinical adoption [111,141]. Concurrently, robust ethical frameworks must address critical issues of data privacy, algorithmic bias, equity in access, and accountability for AI-assisted decisions, ensuring that the march toward precision psychiatry does not compromise fairness or patient autonomy [111].

Beyond ethics and interpretability, the successful translation of this pipeline depends on addressing practical economic considerations and the implementation of best practices. The significant upfront investment required for multi-omics data acquisition, model development, and validation must be justified by demonstrating long-term cost-effectiveness, potentially through reducing ineffective treatment trials and enabling preventative care, as evidenced in health-economic analyses of digital interventions [161]. Furthermore, the adoption of best practices—such as rigorous external validation on diverse cohorts, proactive mitigation of dataset biases, and co-design of tools with end-users to ensure clinical utility—is essential to build reliable, generalizable, and actionable systems. These practices should be guided by evolving reporting standards and frameworks for clinical prediction models [162] to ensure robust integration into real-world healthcare workflows.

### 6.5. Limitations of This Review

This narrative review, while comprehensive, has several inherent limitations. First, its scope, centered on the integrative paradigm of AI, NP, and the gut–brain axis in depression, necessarily means that some adjacent areas (e.g., detailed pharmacogenomics of standard antidepressants, other neuroimmune pathways) are not covered in depth. Second, as a narrative synthesis, it is susceptible to selection bias because the cited literature reflects the authors’ curation of key developments rather than a systematic, exhaustive search of all available evidence. Finally, because the field is rapidly evolving, some of the most recent breakthroughs or clinical trial results may not be captured.

## 7. Conclusions

In conclusion, this review synthesizes and advocates for a fundamental shift in understanding and treating depression—from a reductionist, single-target approach to a dynamic, systems-level framework driven by the convergence of network pharmacology, multi-omics, and artificial intelligence. The gut–brain axis paradigm powerfully demonstrates depression as a network disorder, providing concrete mechanistic insights into biotypes and emphasizing the microbiome as a key therapeutic target. AI acts as an indispensable computational engine, transforming complex data into actionable hypotheses for drug discovery and personalized treatment. Although significant translational challenges remain—such as data integration, model interpretability, and biological validation—they represent critical frontiers for future research rather than fundamental limitations. Moving forward, success depends on fostering interdisciplinary collaboration to develop causality-aware, interpretable, and ethically governed AI tools rooted in diverse longitudinal data. Ultimately, this synergistic approach paves the way for redefining depression therapies by providing precise, mechanism-based, and highly personalized solutions, thereby bridging the gap between complex systems biology and effective clinical care.

## Figures and Tables

**Figure 1 cimb-47-01061-f001:**
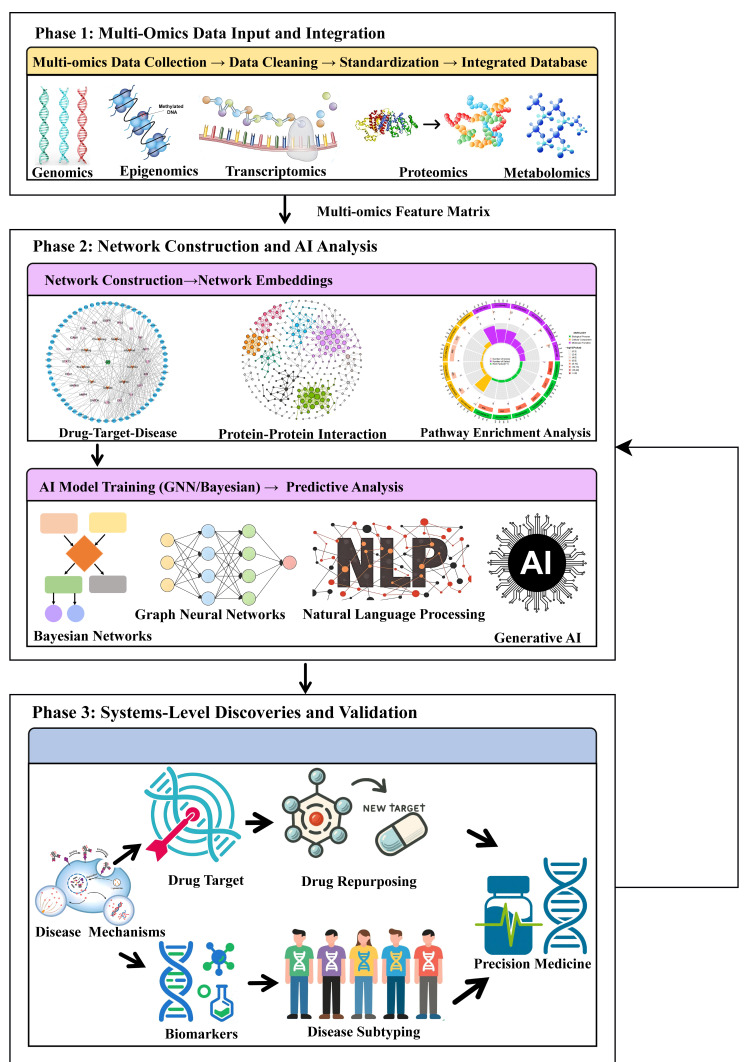
Integrated workflow of network pharmacology and artificial intelligence (AI) for multi-omics analysis in major depressive disorder (MDD). The rightward arrow denotes the primary direction of the workflow and data transmission, while the vertical arrow indicates data transfer or sequential progression among phases. Phase 1 (Multi-omics data input and integration): Raw multi-omics data are collected, cleaned, standardized, and integrated into a unified database. Phase 2 (Network construction and AI analysis): Biological networks (e.g., drug–target–disease and protein–protein interaction) are constructed and converted into numerical embeddings. AI models—including graph neural networks, Bayesian networks, natural language processing, and generative AI—are then trained to generate predictive hypotheses. Phase 3 (systems-level discoveries and validation): Mechanistic insights (including disease mechanisms, novel targets, and biomarkers) inform translational applications such as drug repurposing and disease subtyping, ultimately enabling precision medicine strategies.

**Figure 2 cimb-47-01061-f002:**
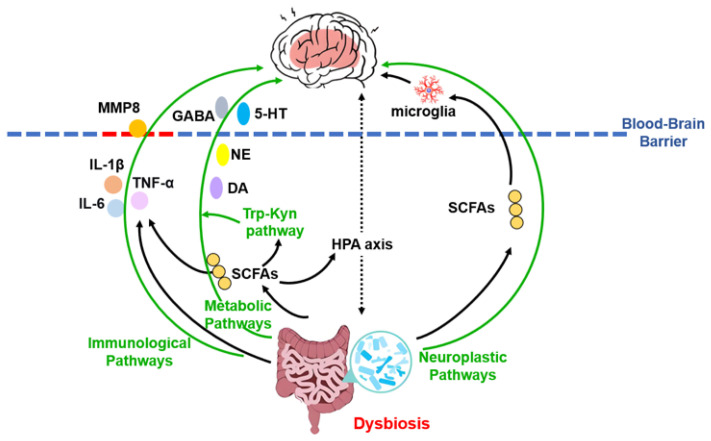
Schematic overview of the microbiota–gut–brain axis mechanisms in depression. This integrative model illustrates how gut dysbiosis contributes to the pathophysiology of major depressive disorder through three core, interacting pathways: neuroimmune, metabolic, and neuroplastic. Solid arrows denote the direction of effects; green arrows highlight the three main mechanistic categories. The blood–brain barrier is represented by a dashed line, with red dashes indicating its compromised, permeable state. Abbreviations: MMP8, matrix metalloproteinase-8; 5-HT, serotonin; DA, dopamine; GABA, γ-aminobutyric acid; HPA, hypothalamic-pituitary-adrenal; IL, interleukin; NE, norepinephrine; SCFAs, short-chain fatty acids; Trp-Kyn, tryptophan-kynurenine.

**Table 1 cimb-47-01061-t001:** Multi-omics technologies and their contributions to MDD research.

Omics Layer	Technology and Purpose	Key Findings/Candidate Biomarkers	Exemplary Study (Findings)	Implications for MDD Research
Genomics	GWAS; to identify inherited risk loci. Polygene Risk Score.	102 independent risk variants; 269 implicated genes (e.g., related to neuronal development).	Howard et al. [18]: Large-scale meta-analysis.	Quantifies genetic susceptibility; foundation for understanding heritability.
Epigenomics	DNA methylation/hydroxymethylation and histone modifications link the environment to gene expression.	*NR3C1* gene methylation mediates the effects of early life stress on symptom severity.	Efstathopoulos et al. [21]: Salivary DNA in adolescents.	Illustrates how experience “programs” gene expression; a mechanistic bridge.
Transcriptomics	RNA-seq, snRNA-seq, and other methods profile functional gene activity at bulk or single-cell resolution.	Cell-type-specific dysregulation (e.g., deep-layer excitatory neurons, oligodendrocyte precursors).	Nagy et al. [23]: snRNA-seq on dlPFC.	Identifies key dysregulated cell populations; pinpoints precise pathological targets.
Proteomics	Large-scale protein profiling identifies functional effectors and treatment-response markers.	Depression-associated proteins (immune, cell communication). Inflammatory proteins change post-treatment.	Bot et al. [25]; Liu et al. Treatment-response studies [26].	Aids biomarker discovery and predicts/personalizes therapeutic response.
Metabolomics	Profiling small molecules; terminal readout of systemic physiology.	Perturbed purine/fatty acid metabolism; altered amino acids (e.g., glutamate, glycine); inosine as a potential biomarker.	Studies on young MDD patients [28,29].	Reflects functional output; offers diagnostic/severity stratification markers.
Integrative Multi-Omics	Computational integration of multiple omics layers (genomics, epigenomics, transcriptomics, proteomics, metabolomics), often with metagenomics.	Holistic causal networks spanning from genetic variation to metabolic output elucidate the MGB axis.	(To be discussed in later chapters)	Aims to decode disease heterogeneity, delineate molecular subtypes, and elucidate the mechanistic interplay between genetic predisposition, environmental triggers, and dysregulated pathways.

**Table 2 cimb-47-01061-t002:** Key artificial intelligence and computational approaches for integrating multi-omics data in depression research.

Core Integrative Challenge	AI/Computational Approach	Primary Task and Role in Integration	Exemplary Data Inputs	Representative Application (Study Aim)	Key References
Deciphering Disease Heterogeneity	Unsupervised Learning (e.g., Clustering)	Identifying data-driven biotypes or subgroups that transcend symptom-based diagnoses, linking them to distinct biological mechanisms.	Multi-omics profiles, neuroimaging data (fMRI, EEG), and clinical phenotype arrays.	Discovery of neurophysiological or molecular subtypes with differential treatment responses.	[49]
Multimodal Feature Fusion and Prediction	Supervised Learning (e.g., Support Vector Machines, Random Forest)	Integrating diverse features across omics layers to build diagnostic, prognostic, or treatment response models; identifying salient biomarkers.	Genomics, epigenomics (methylation), metabolomics, clinical, and demographic variables.	Predicting antidepressant response using combined genetic, epigenetic, and clinical data.	[50,51]
Modeling Complex Biological Systems	Graph Neural Networks (GNNs), Dynamic Bayesian Networks.	Inferring key hubs and dysregulated interactions within biological networks (e.g., brain connectomes, molecular pathways); modeling temporal dynamics.	Neuroimaging-derived connectomes, protein–protein interaction networks, and longitudinal omics data.	Mapping aberrant functional connectivity in depression or inferring context-specific gene regulatory networks.	[52]
Bridging Subjective Experience with Biology	Natural Language Processing (NLP)	Quantifying subjective experience and clinical narrative into analyzable digital phenotypes, enabling correlation with biological data.	Electronic health records, patient-generated text, transcribed interviews.	Correlating linguistic markers from therapy sessions with neuroimaging biomarkers to predict treatment outcomes.	[53,54,55,56]
Generating Hypotheses and Personalizing Intervention	Generative AI (e.g., Large Language Models)	Aiding in hypothesis generation, synthesizing patient data for personalized insights, and providing scalable digital therapeutic support.	Unstructured clinical notes, patient self-reports, multimodal health data.	AI-assisted journaling for therapy personalization and support in treatment-resistant depression.	[57,58,59,60,61,62,63]

**Table 3 cimb-47-01061-t003:** Natural products targeting the microbiota–gut–brain (MGB) axis.

Class of Compound	Representative Compound	Key Mechanisms Related to MGB Axis	Primary Effect	References
Flavonoids	Quercetin	Modulates NMDA/GABA; downregulates TNF-α, IL-1β; activates Nrf-2; influences BDNF, PI3K/Akt, MAPK/ERK.	Multi-target neuroprotection	[117]
	Puerarin	Restores gut microbiota; reduces Desulfovibrio; enhances BDNF, IκB-α; suppresses NF-κB.	Antidepressant effect in CUMS rats	[119]
Alkaloids	Berberine	Elevates Firmicutes; reduces Bacteroides; alters SCFAs (increasing isovaleric acid).	Activates MGB axis; elevated hippocampal BDNF/monoamines; alleviates behaviors	[121]
Polysaccharides	*Schisandra chinensis* polysaccharide	Inhibits HPA axis; reduces oxidative stress; modulates gut microbiota.	Ameliorates depressive behaviors; restores neuroplasticity	[123]
	*Corydalis yanhusuo* polysaccharides	Regulates gut microbiota and SCFAs; upregulate *TPH-2*.	Enhances monoamines (Norepinephrine, Dopamine, 5-HT) and BDNF	[124]
Quinoid Diterpenes	Cryptotanshinone	Attenuates inflammation; rebalances microbiota (e.g., reduces *Parabacteroides merdae*); modulates PI3K-AKT	Alleviates depression-like phenotypes	[125]
Polyphenol Extract	Apple Polyphenol Extract	Restores microbial homeostasis; inhibits NF-κB; enhances occludin, ZO-1; normalizes HPA axis	Contributes to antidepressant efficacy	[126]
Oligosaccharide Esters	Polygalae Radix Oligosaccharide Esters	Increases beneficial bacteria; reinforces gut barrier; modulates Trp-Kyn pathway	Regulates brain neurotransmission	[127]
Extract	*Cuscutae Semen* extract	Modulates GM; inhibits NLRP3/NF-κB pathway; preserves synaptic integrity	Antidepressant and anti-inflammatory activities	[128]
	Saffron Extract	Increases beneficial gut microbiota (e.g., *Akkermansia*, *Muribaculaceae flora*); reduces neurotoxic dimethylamine; modulates brain proteomics	Exerts antidepressant effects	[129]

**Table 4 cimb-47-01061-t004:** Translational applications of artificial intelligence in depression management.

Application Domain	Core Tasks	Data Types	Representative Methods	References
Biomarker Discovery and Validation	1. To integrate multi-omics data for constructing biosignatures capable of diagnostic and prognostic stratification. 2. To identify key biological pathways associated with distinct depression biotypes (e.g., inflammatory or metabolic subtypes).	Genomics, proteomics, metabolomics, gut metagenomic data, and clinical records.	Multi-omics integration (e.g., DIABLO); supervised learning models (for classification and prediction).	[17,76]
Drug Discovery and Repurposing	1. To predict novel drug–disease associations and potential therapeutic targets. 2. To identify multi-target therapies and synergistic drug combinations addressing disease complexity.	Drug–target interaction networks, gene-expression profiles, and electronic health records.	Graph neural networks, deep-learning models (e.g., DeepDRA), knowledge-graphs, or tensor-based approaches.	[109,110,111,112]
Precision Medicine Framework	1. To develop multi-omics-based patient stratification models that define biologically distinct “biotypes.” 2. To predict individualized treatment response and match interventions (e.g., anti-inflammatory nutraceuticals, prebiotics) to specific biotypes. 3. To construct “digital twin” or dynamic in silico models that simulate disease progression and virtually test interventions, generating prioritized hypotheses for clinical decision-making.	Gut metagenomic, plasma metabolomic, immunomic, genomic data; clinical phenotypes and treatment-outcome data.	Multivariate predictive modeling, machine learning classifiers, and hybrid modeling combining mechanistic and AI-based approaches.	[42,80,126,127,130,131,132,49]

**Table 5 cimb-47-01061-t005:** Psychobiotic and dietary interventions for depression: mechanisms and evidence base.

Intervention Category	Specific Examples	Primary Mechanisms of Action	Evidence Base and Effects
Probiotics	*Lactobacillus rhamnosus* JB-1 *Bifidobacterium* spp. *Lactobacillus helveticus*	1. Modulates the HPA axis, reducing stress response. 2. Decreases pro-inflammatory cytokines (e.g., IL-1β, TNF-α). 3. Influences neurotransmitters (e.g., increases GABA, modulates glutamate) and elevates neurotrophic factors (e.g., BDNF). 4. Enhances intestinal barrier function, reducing inflammation.	Preclinical: Significantly reduces depressive- and anxiety-like behaviors in rodents (e.g., JB-1 strain); normalizes HPA axis hyperactivity and hippocampal BDNF expression [150,151,152,153]. Clinical: Multiple human trials show specific probiotic formulations can improve depressive and anxiety symptoms and related biochemical markers, though optimal strains and treatment duration require further validation [148,149,150].
Prebiotics	Galacto-oligosaccharides (GOS) Resistant Starch (RS) (Found in bananas, whole grains, garlic, etc.)	1. Selectively promotes growth of beneficial bacteria, improving gut microbiota composition [154]. 2. Modulates immune and neuroendocrine functions via microbial metabolites (e.g., short-chain fatty acids) [154]. 3. Possesses anti-inflammatory and antioxidant properties (e.g., cocoa flavanols) [157].	Preclinical: GOS and RS alleviate stress-induced anxiety and depressive behaviors in animal models [155,156]. Clinical: Preliminary studies indicate certain prebiotic supplements improve subjective stress response and mood; however, large-scale targeted trials for depression are still needed [154,157].
Dietary Patterns	Mediterranean Diet (rich in olive oil, fruits, vegetables, fermented dairy) Very-Low-Carbohydrate Ketogenic Diet (VLCKD) Calorie Restriction	1. Mediterranean Diet: Provides abundant fiber and polyphenols, shaping a beneficial microbiota, systemically attenuating oxidative stress and inflammation [158]. 2. Ketogenic Diet: Ketone bodies may influence neuronal excitability, mitochondrial function, and gut microbiota, showing potential for treatment-resistant depression [159]. 3. Calorie Restriction: Potential antidepressant effects via weight loss, reduced inflammation, and microbiota changes [160].	Observational: Adherence to a Mediterranean dietary pattern is associated with a lower risk of depression [158]. Interventional: Ketogenic diets show mood-improving effects in some patients with comorbid epilepsy or metabolic syndrome; evidence for calorie restriction on mood is mixed, requiring more clinical data [159,160].
Fecal Microbiota Transplantation (FMT)	Transplantation of gut microbiota from a healthy donor to a recipient.	1. Directly and holistically reshapes the recipient’s gut microbial ecosystem. 2. Restores microbial diversity and function, correcting depression-associated dysbiosis. 3. Indirectly influences brain function via gut–brain axis pathways (immune, neural, metabolic).	Preclinical: Transplanting microbiota from depressed patients into germ-free mice induces depressive-like behaviors, whereas Transplantation of healthy microbiota is protective. Clinical: Currently primarily used for gastrointestinal disorders. Preliminary case reports/small-scale studies in depression show potential, but the approach remains in early exploration, requiring rigorous Randomized Controlled Trials to validate safety and efficacy [147].

## Data Availability

No new data were created or analyzed in this study. Data sharing is not applicable to this article.

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
