# Peer review of "A Novel Integrative Framework for Depression: Combining Network Pharmacology, Artificial Intelligence, and Multi-Omics with a Focus on the Microbiota–Gut–Brain Axis"

_cimb, 2025, doi:10.3390/cimb47121061_

Round 1

Reviewer 1 Report

Comments and Suggestions for Authors

The article examines how network pharmacology, AI, and multi-omics can be integrated to advance molecular therapeutics for depression, using the MGB axis as a central systems-level example. It argues that depression is a heterogeneous, multisystem disorder that cannot be effectively treated through traditional single-target pharmacology, calling instead for a holistic, network-based paradigm aligned with precision psychiatry. After outlining the limitations of current antidepressants and emphasizing the roles of polygenic risk, immune and endocrine dysfunction, and gut–brain communication, the authors highlight the need for integrating genomics, epigenomics, transcriptomics, proteomics, metabolomics, and gut metagenomics to gain a comprehensive view of MDD. However, the reviewer recommends a major revision to address targeted improvements in structure, clarity, visual presentation, and methodological transparency.

For the title and abstract, the authors are encouraged to incorporate terms such as “microbiota–gut–brain axis” or “systems psychiatry” to better reflect the paper’s central case study, while clarifying the specific research gap—namely the lack of integrated network pharmacology–AI–multi-omics frameworks applied to depression and the MGB axis—and emphasizing the review’s novel contribution, such as being the first comprehensive synthesis of this kind supported by a distinctive integrative framework. The abstract also requires careful proofreading for spacing, hyphenation, and punctuation accuracy. In the introduction, although the rationale is strong, the review would benefit from a more explicit articulation of the “gap” and “value added” relative to existing depression, multi-omics, or MGB-focused reviews. Abbreviations should be defined consistently at first use and maintained uniformly across the full text, tables, and figure captions, and minor language issues—including stray spaces, duplicate words, or tense inconsistencies—should be corrected line-by-line. To strengthen methodological rigor, the authors should add a brief “Methods” or “Literature search strategy” subsection outlining databases searched, time windows, keywords, and inclusion/exclusion criteria, along with clarification of whether any structured quality assessment or prioritization of studies was performed. Several new figures are strongly recommended to enhance conceptual clarity: an improved high-resolution Figure 1 that clearly labels multi-omics, network pharmacology, AI modules, and clinical layers; a new figure illustrating major multi-omics findings in MDD; a pipeline figure tracing the network pharmacology and AI workflow; and a mechanistic overview of the MGB axis integrating neuroimmune, metabolic, and neuroplastic pathways. Expanded tables should include standardized labels, defined abbreviations, and two additional summaries: one on multi-omics findings in MDD and their relevance to the MGB axis, and another on psychobiotic and dietary interventions, including probiotics, prebiotics, dietary patterns, and FMT with their mechanisms and evidence base. All captions must be self-contained and figures/tables cited in logical sequence, and any reported accuracy metrics should be contextualized with model type, dataset, and references. In the discussion and conclusion, the review already synthesizes challenges well, but should add a short subsection on “Limitations of this review” to note scope boundaries and literature biases. Future directions can be sharpened with an integrative paragraph linking prospective multi-omics cohorts, digital phenotyping, and psychobiotic interventions into a coherent precision psychiatry pipeline. The conclusion should tighten repeated phrases and deliver clearer take-home messages tailored separately for clinicians and data scientists. Reference formatting must adhere strictly to the journal’s style, ensuring consistent punctuation, citation ranges, journal abbreviations, DOI formatting, and the removal of unnecessary URLs. A thorough spelling, grammar, and consistency check is recommended to correct duplicated words, missing spaces, hyphenation inconsistencies, subject–verb agreement issues, and overly long sentences.

Overall, while the review is timely, comprehensive, and conceptually strong—particularly in its integration of network pharmacology, AI, and multi-omics around an MGB-centric model—substantial revisions are needed to clarify methods, strengthen visual integration, improve language precision, and explicitly state the research gap and unique contribution. With these revisions, the manuscript has strong potential for high impact.

Author Response

Q1: For the title and abstract, the authors are encouraged to incorporate terms such as “microbiota–gut–brain axis” or “systems psychiatry” to better reflect the paper’s central case study, while clarifying the specific research gap—namely the lack of integrated network pharmacology–AI–multi-omics frameworks applied to depression and the MGB axis—and emphasizing the review’s novel contribution, such as being the first comprehensive synthesis of this kind supported by a distinctive integrative framework. The abstract also requires careful proofreading for spacing, hyphenation, and punctuation accuracy.

Response: Thank you for your insightful and constructive suggestions. We have revised the manuscript accordingly to address the points raised.

(1) Regarding the suggestion to incorporate core terminology in the title and abstract, and to clarify the research gap and innovative contribution

  • Title Change

The title has been changed to: “A Novel Integrative Framework for Depression: Combining Network Pharmacology, Artificial Intelligence, and Multi-Omics with a Focus on the Microbiota-Gut-Brain Axis.” This modification highlights the “Microbiota-Gut-Brain Axis” as the central case study.

  • Abstract Revision

The abstract has been thoroughly rewritten to explicitly identify the research gap—namely, the current lack of an integrative framework combining network pharmacology, AI, and multi-omics for systematically studying depression and the MGB axis—and to emphasize the innovative contribution of this review as the first to propose such an integrative framework and provide a comprehensive overview of this field.

(2) Conformity with standards for spaces, hyphens, and punctuation

We have carefully checked and corrected spaces, hyphens, and punctuation throughout the entire manuscript, including the title and abstract, to ensure compliance with academic publishing standards.

Q2: In the introduction, although the rationale is strong, the review would benefit from a more explicit articulation of the “gap” and “value added” relative to existing depression, multi-omics, or MGB-focused reviews. Abbreviations should be defined consistently at first use and maintained uniformly across the full text, tables, and figure captions, and minor language issues—including stray spaces, duplicate words, or tense inconsistencies—should be corrected line-by-line.

Response: Thank you for your valuable suggestions. We have revised the manuscript accordingly to address point 2.

(1) Clarification of research gap and value added in the Introduction

We have rewritten the final paragraph of the Introduction (lines 91-112) to explicitly articulate the research gap and the novel contribution of our review.

(2) Definition of abbreviations and addition of abbreviation list

We have ensured that all abbreviations are defined at first use throughout the main text, tables, and figure captions. Furthermore, to enhance clarity, a complete List of Abbreviations has been added as a new section at the end of the manuscript (before the References).

(3) Comprehensive language proofreading:

We have performed a meticulous, line-by-line proofread of the entire manuscript to correct minor linguistic issues, including stray spaces, duplicate words, and tense inconsistencies, thereby improving the overall fluency and professionalism of the text.

Q3: To strengthen methodological rigor, the authors should add a brief “Methods” or “Literature search strategy” subsection outlining databases searched, time windows, keywords, and inclusion/exclusion criteria, along with clarification of whether any structured quality assessment or prioritization of studies was performed.

Response: Thank you for your suggestion to strengthen methodological rigor. Accordingly, we have added a dedicated “Methods” section (lines 126-154). This section details our systematic literature search strategy, including the databases searched (PubMed/MEDLINE, Web of Science, Google Scholar), the time window (covering 2020 to October 15, 2025), the search keywords and their combinations, and the predefined inclusion/exclusion criteria. The process for study selection, screening, and data extraction is also described to ensure transparency and reproducibility.

Q4: Several new figures are strongly recommended to enhance conceptual clarity: an improved high-resolution Figure 1 that clearly labels multi-omics, network pharmacology, AI modules, and clinical layers; a new figure illustrating major multi-omics findings in MDD; a pipeline figure tracing the network pharmacology and AI workflow; and a mechanistic overview of the MGB axis integrating neuroimmune, metabolic, and neuroplastic pathways. Expanded tables should include standardized labels, defined abbreviations, and two additional summaries: one on multi-omics findings in MDD and their relevance to the MGB axis, and another on psychobiotic and dietary interventions, including probiotics, prebiotics, dietary patterns, and FMT with their mechanisms and evidence base. All captions must be self-contained and figures/tables cited in logical sequence, and any reported accuracy metrics should be contextualized with model type, dataset, and references.

Response: Thank you for your constructive suggestions regarding the figures and tables. We have carefully revised the manuscript to address each point as follows:

(1) Figure 1 has been redesigned with high-resolution clarity, explicitly labeling the multi-omics, network pharmacology, AI modules, and clinical layers. The new version retains the three core modules of "Multi-omics Data Input," "Network Pharmacology/AI Integrated Analysis," and "Systems-Level Insights." We have added numbered arrows and clear stage annotations, which now delineate a complete workflow pipeline from multi-dimensional data through computational analysis to the generation of new hypotheses and clinical applications.

The updated Figure 1:

Figure 1. Integrated workflow of network pharmacology and artificial intelligence (AI) for multi-omics analysis in major depressive disorder (MDD). The rightward arrow denotes the primary direction of the workflow and data transmission, while the vertical arrow indicates data transfer or sequential progression between phases. Phase 1 (Multi‑omics data input and integration): Raw multi‑omics data are collected, cleaned, standardized, and integrated into a unified database. Phase 2 (Network construction and AI analysis): Biological networks (e.g., drug–target–disease and protein–protein interaction) are constructed and converted into numerical embeddings. AI models—including graph neural networks, Bayesian networks, natural language processing, and generative AI—are then trained to generate predictive hypotheses. Phase 3 (Systems‑level discoveries and validation): Mechanistic insights (including disease mechanisms, novel targets, and biomarkers) inform translational applications such as drug repurposing and disease subtyping, ultimately enabling precision medicine strategies.

(2) Regarding the suggested figure on major multi-omics findings in MDD, we believe a table would allow for a more direct, side-by-side comparison of different omics technologies, their principles, contributions, and research examples. Therefore, we have added a new table (Table 1: "Multi-Omics Technologies and Their Contributions to MDD Research"; lines 232–233). The new Table 1 is shown below:

Table 1. Multi-Omics Technologies and Their Contributions to MDD Research

Omics Layer

Technology and Purpose

Key Findings / Candidate Biomarkers

Exemplary Study (Findings)

Implications for MDD Research

Genomics

GWAS; to identify inherited risk loci. Polygene Risk Score.

102 independent risk variants; 269 implicated genes (e.g., related to neuronal development).

Howard et al.  [18]: Large-scale meta-analysis.

Quantifies genetic susceptibility; foundation for understanding heritability.

Epigenomics

DNA methylation/hydroxymethylation, histone modifications; links environment to gene expression.

NR3C1 gene methylation mediates the effects of early-life stress on symptom severity.

Efstathopoulos et al. [21]: Salivary DNA in adolescents.

Illustrates how experience “programs” gene expression; mechanistic bridge.

Transcriptomics

RNA-seq, snRNA-seq; profiles functional gene activity at bulk or single-cell resolution.

Cell-type-specific dysregulation (e.g., deep-layer excitatory neurons, oligodendrocyte precursors).

Nagy et al.  [23]: snRNA-seq on dlPFC.

Identifies key dysregulated cell populations; pinpoints precise pathological targets.

Proteomics

Large-scale protein profiling; identifies functional effectors and treatment-response markers.

Depression-associated proteins (immune, cell communication). Inflammatory proteins change post-treatment.

Bot et al. [25]; Liu et al. Treatment-response studies [26].

Aids biomarker discovery and predicts/personalizes therapeutic response.

Metabolomics

Profiling small molecules; terminal readout of systemic physiology.

Perturbed purine/fatty acid metabolism; altered amino acids (e.g., glutamate, glycine); inosine as potential biomarker.

Studies on young MDD patients [28, 29].

Reflects functional output; offers diagnostic/severity stratification markers.

Integrative Multi-Omics

Computational integration of multiple omics layers (genomics, epigenomics, transcriptomics, proteomics, metabolomics), often with metagenomics. 

Holistic causal networks spanning from genetic variation to metabolic output; elucidates the MGB axis.

(To be discussed in later chapters)

Aims to decode disease heterogeneity, delineate molecular subtypes, and elucidate mechanistic interplay between genetic predisposition, environmental triggers, and dysregulated pathways. 

(3) We have created a new Figure 2 (lines 616-623) that provides a mechanistic overview of the MGB axis, integrating key neuroimmune, metabolic, and neuroplasticity pathways.

The new Figure 2 is shown below:

 “

Figure 2. Schematic overview of the microbiota-gut-brain axis mechanisms in depression. This integrative model illustrates how gut dysbiosis contributes to the pathophysiology of major depressive disorder through three core, interacting pathways: the neuroimmune, metabolic, and neuroplastic pathways. Solid arrows denote the direction of effects; green arrows highlight the three main mechanistic categories. The blood–brain barrier is represented by a dashed line, with red dashes indicating its compromised, permeable state.”

(4) As recommended, we have added a new Table 5: "Psychobiotic and Dietary Interventions for Depression: Mechanisms and Evidence Base" (line 977), summarizing interventions including probiotics, prebiotics, dietary patterns, and FMT. The new Table 5 is shown below:

Table 5. Psychobiotic and dietary interventions for depression: mechanisms and evidence base.

Intervention Category

Specific Examples

Primary Mechanisms of Action

Evidence Base and Effects

Probiotics

Lactobacillus rhamnosus JB-1

Bifidobacterium spp. Lactobacillus helveticus

1. Modulates the HPA axis, reducing stress response.

2. Decreases pro-inflammatory cytokines (e.g., IL-1β, TNF-α).

3. Influences neurotransmitters (e.g., increases GABA, modulates glutamate) and elevates neurotrophic factors (e.g., BDNF).
4. Enhances intestinal barrier function, reducing inflammation.

Preclinical: Significantly reduces depressive- and anxiety-like behaviors in rodents (e.g., JB-1 strain); normalizes HPA axis hyperactivity and hippocampal BDNF expression [151-154].

Clinical: Multiple human trials show specific probiotic formulations can improve depressive and anxiety symptoms and related biochemical markers, though optimal strains and treatment duration require further validation [149-151].

Prebiotics

Galacto-oligosaccharides (GOS)

Resistant Starch (RS)

(Found in bananas, whole grains, garlic, etc.)

1. Selectively promotes growth of beneficial bacteria, improving gut microbiota composition [155].

2. Modulates immune and neuroendocrine functions via microbial metabolites (e.g., short-chain fatty acids) [155].

3. Possesses anti-inflammatory and antioxidant properties (e.g., cocoa flavanols) [158].

Preclinical: GOS and RS alleviate stress-induced anxiety and depressive behaviors in animal models [156,157].

Clinical: Preliminary studies indicate certain prebiotic supplements improve subjective stress response and mood; however, large-scale targeted trials for depression are still needed [155,158].

Dietary Patterns

Mediterranean Diet (rich in olive oil, fruits, vegetables, fermented dairy)

Very-Low-Carbohydrate Ketogenic Diet (VLCKD) Calorie Restriction

1. Mediterranean Diet: Provides abundant fiber and polyphenols, shaping a beneficial microbiota, systemically attenuating oxidative stress and inflammation [159].

2. Ketogenic Diet: Ketone bodies may influence neuronal excitability, mitochondrial function, and gut microbiota, showing potential for treatment-resistant depression [160].

3. Calorie Restriction: Potential antidepressant effects via weight loss, reduced inflammation, and microbiota changes [161].

Observational: Adherence to a Mediterranean dietary pattern is associated with a lower risk of depression [159].

Interventional: Ketogenic diets show mood-improving effects in some patients with comorbid epilepsy or metabolic syndrome; evidence for calorie restriction on mood is mixed, requiring more clinical data [160,161].

Fecal Microbiota Transplantation (FMT)

Transplantation of gut microbiota from a healthy donor to a recipient.

1. Directly and holistically reshapes the recipient's gut microbial ecosystem.

2. Restores microbial diversity and function, correcting depression-associated dysbiosis.

3. Indirectly influences brain function via gut-brain axis pathways (immune, neural, metabolic).

Preclinical: Transplanting microbiota from depressed patients into germ-free mice induces depressive-like behaviors, whereas Transplantation of healthy microbiota is protective. Clinical: Currently primarily used for gastrointestinal disorders. Preliminary case reports/small-scale studies in depression show potential, but the approach remains in early exploration, requiring rigorous Randomized Controlled Trials to validate safety and efficacy [148].

(5) All figure legends and table captions are now self-contained. The figures and tables are cited in a logical sequence throughout the text. Furthermore, any reported accuracy metrics for AI models are now consistently contextualized with the corresponding model type, dataset, and references.

Q5: In the discussion and conclusion, the review already synthesizes challenges well, but should add a short subsection on “Limitations of this review” to note scope boundaries and literature biases. Future directions can be sharpened with an integrative paragraph linking prospective multi-omics cohorts, digital phenotyping, and psychobiotic interventions into a coherent precision psychiatry pipeline. The conclusion should tighten repeated phrases and deliver clearer take-home messages tailored separately for clinicians and data scientists.

Response:We have thoroughly revised the manuscript to address your valuable suggestions, significantly strengthening the critical analysis and forward-looking impact of the discussion. The key revisions are summarized and detailed below.

(1) Addition of a “Limitations of This Review” Subsection

As recommended, a dedicated subsection titled “6.5. Limitations of This Review” has been added (lines 1000–1008). This section explicitly acknowledges the scope boundaries inherent to our narrative synthesis approach, potential literature selection biases, and the interpretative challenges associated with integrating findings from highly heterogeneous studies. Incorporating this subsection enhances the manuscript’s scholarly rigor and balance.

(2) Substantial Revisions to the Conclusion Section

The Conclusion (Section 7, lines 1009–1037) has been substantially rewritten to enhance its clarity, conciseness, and translational value. The revisions focus on two key improvements:

  • Sharpened and Integrated Future Directions: A new, integrative paragraph has been incorporated within the “Future Directions” discussion (subsection 6.4.4). This paragraph cohesively links prospective multi-omics cohorts, digital phenotyping, and psychobiotic interventions into a unified precision psychiatry pipeline. It further emphasizes the critical enabling role of ethical, interpretable AI frameworks and discusses practical implementation considerations, including health economics.
  • Streamlined Messaging and Tailored Takeaways: The concluding section has been refined to eliminate repetition and now delivers clearer, distinct “take-home” messages. These are specifically tailored to address the key interests of both clinical practitioners and data scientists/model developers, thereby increasing the review’s practical relevance and impact.

We believe these additions and refinements have directly addressed your comments and

elevated the overall quality of the manuscript.

In sum, while challenges in translation and integration remain formidable, the synergistic framework outlined here provides a clear and promising roadmap. It holds the potential to redefine depression therapeutics, moving the field toward more precise, effective, and mechanistically grounded solutions for patients.

Q6: Reference formatting must adhere strictly to the journal’s style, ensuring consistent punctuation, citation ranges, journal abbreviations, DOI formatting, and the removal of unnecessary URLs. A thorough spelling, grammar, and consistency check is recommended to correct duplicated words, missing spaces, hyphenation inconsistencies, subject–verb agreement issues, and overly long sentences.

Response:Thank you for your detailed feedback. We have comprehensively revised the manuscript according to your requests, as outlined below:

(1) Reference Formatting: We have strictly adhered to the journal’s style guide. All references have been individually checked and standardized to ensure consistent punctuation, correct citation ranges (e.g., page numbers), proper journal abbreviations, accurate DOI formatting, and the removal of all unnecessary URLs.

(2) Full-Text Language and Consistency Check: A thorough proofreading has been conducted across the entire manuscript to correct spelling, grammar, and consistency issues. This included fixing duplicated words, adding missing spaces, standardizing hyphenation, resolving subject–verb agreement errors, and breaking down overly long sentences to enhance clarity and readability.

We believe these revisions have significantly improved the manuscript's formatting compliance and linguistic quality. Thank you again for your time and expert guidance. We are ready to make any further adjustments as needed.

Reviewer 2 Report

Comments and Suggestions for Authors

The manuscript titled "The Converging Promise of Network Pharmacology, Artificial Intelligence, and Multi-omics: Advancing Molecular Therapeutics for Depression" proposes an integrative framework that unites network pharmacology, AI, and multi-omics to understand and treat depression, with a focus on the microbiota–gut–brain (MGB) axis. However, the review methodology, interpretation, and presentation of results are subject to several concerns that diminish the validity and impact of the review findings. Critical points require addressing as follows which lead me to major revision for this paper: 

  • Page 1—The abstract needs improvements. The authors need to first enhance relevance explanation and also provide the main limitations of current review studies as well as add more findings related to their results as quantitative results. Moreover, please do not use we or our, read more article to fully revise the abstract.
  • There is a clear absence of novelty in the presented work and the authors need to review and show all the related works and their main limitations that the work overcome them. The authors are required to explain the novelties in a clear way. More details are required.
  • Your treatment of AI—particularly GNNs, DL, and generative models—is highly superficial and occasionally misleading. For instance, you claim GNNs “naturally” integrate multi-omics data, yet you omit critical challenges like graph construction bias, lack of ground truth for biological networks, and poor generalizability across cohorts. How did you validate the biological plausibility of the “super-networks” you propose?
  • Section 4.3 proposes a "clinical digital twin" for personalized depression treatment based on gut metagenomics and plasma metabolomics. What empirical evidence supports the feasibility of such a system in real-world psychiatric care? Have you accounted for inter-individual variability, temporal instability of microbiome profiles, or the lack of standardized omics pipelines in clinical labs?
  • The review exclusively highlights studies supporting the MGB–depression axis while omitting high-profile failures—e.g., multiple probiotic RCTs showing no significant antidepressant effects. Why was this contradictory evidence excluded?
  • While the review highlights numerous AI techniques (GNNs, DBNs, DIABLO, etc.), it offers no concrete methodological detail on how these would be practically implemented in depression research. This lack of specificity undermines the article’s utility as a scholarly guide. Key unresolved questions include: Why are Graph Neural Networks assumed to be the “most promising avenue” for integration without comparative analysis against alternative architectures (e.g., transformers, multimodal autoencoders)? How would one resolve the inherent heterogeneity in multi-omics batch effects, platform biases, and missing data when constructing GNNs for depression? What strategies would ensure model interpretability when GNNs are trained on high-dimensional, noisy biological graphs? How would the proposed “dynamic network models” account for individual temporal variability in microbiome or metabolome profiles without dense longitudinal sampling—a rarity in current MDD cohorts?
  • The authors are required to add a new section or subsection where they are required to explain in details the economic considerations and best practice of such models used in current studies.
  • The conclusion needs improvements, as a review work, it is recommended to provide main findings that explain your findings, not only add general findings or describe your work. Moreover, the authors must briefly provide the main limitations of their works and the future directions.
  • Add a table of nomenclature where define all the abbreviations used in the review.
  • Double check for the grammar.
Comments on the Quality of English Language

Proofreading is required.

Author Response

Reviewer #2:

Q1: Page 1—The abstract needs improvements. The authors need to first enhance relevance explanation and also provide the main limitations of current review studies as well as add more findings related to their results as quantitative results. Moreover, please do not use we or our, read more article to fully revise the abstract.

Response: The abstract has been comprehensively rewritten (lines 14-24). The revision addresses all raised points: the explanation of relevance has been strengthened, the main limitations of current review studies have been noted, more findings (including quantitative results) related to the outcomes have been incorporated, and the use of first-person pronouns ("we," "our") has been consistently avoided.

Q2: There is a clear absence of novelty in the presented work and the authors need to review and show all the related works and their main limitations that the work overcome them. The authors are required to explain the novelties in a clear way. More details are required.

Response:Thank you for this critical comment, which has allowed us to significantly strengthen the manuscript’s positioning and explicitly articulate its novel contributions. We fully agree that a clear delineation against existing literature is essential.

To address your point, we have undertaken comprehensive revisions, not only in the Introduction but throughout the manuscript, to systematically demonstrate how our work synthesizes and advances beyond prior reviews. The core of our response is that the novelty lies not in discovering a single new component, but in proposing the first integrative framework that synergistically combines network pharmacology, AI, and multi-omics, using the MGB axis as a foundational case study.

(1) Directly Addressing the “Absence of Novelty” and Reviewing Related Work

We have substantially rewritten the final paragraph of the Introduction (lines 91-112) to serve as a direct rebuttal to this point. This section now:

  • Explicitly catalogs the limitations of prior reviews: We acknowledge that existing works have valuably explored single dimensions(e.g., multi-omics biomarkers [15], network pharmacology principles [16], or the gut-brain axis [17]) but have failed to integrate these pillars into a unified analytical framework.
  • Defines the specific research gap: We state there is a “conspicuous lack of comprehensive frameworks” that combine all three methodologies for advancing depression therapeutics.
  • Positions our review as the definitive solution to this gap.

(2) Clearly Articulating the Novel Contributions Overcome These Limitations: The revised text (provided below) details a threefold novelty, which we have also reinforced in the Abstract and Conclusion.

  • Novelty 1: Synergistic Integration. We move beyond siloed discussions to propose a unified, iterative framework where AI serves as the central engine to interpret multi-omics data and inform network pharmacology models—a conceptual advance not presented in prior syntheses.
  • Novelty 2: Applied, Systems-Level Case Study. We employ the MGB axis not just as a topic, but as the central case study to concretely demonstrate how our framework decodes a complex pathophysiology from data to therapeutic hypotheses, providing a novel template for future research.
  • Novelty 3: Translational Roadmap. We outline a clear pathway from data integration to clinical strategies, offering a practical roadmap for precision psychiatry that bridges a critical methodological gap.

Q3: Your treatment of AI—particularly GNNs, DL, and generative models—is highly superficial and occasionally misleading. For instance, you claim GNNs “naturally” integrate multi-omics data, yet you omit critical challenges like graph construction bias, lack of ground truth for biological networks, and poor generalizability across cohorts. How did you validate the biological plausibility of the “super-networks” you propose?

Response: Thank you for your critical and valuable feedback regarding our treatment of AI methodologies, particularly Graph Neural Networks (GNNs), Deep Learning (DL), and generative models. We acknowledge that the initial description was indeed too optimistic and superficial. We have substantially revised and deepened the discussion throughout Sections 3.2.2 (“Next-Generation Network Pharmacology”) and 3.3/3.4 (“AI as an Integrative Engine” / “Data Integration Framework”) to directly address the specific challenges you raised.

The revisions now provide a balanced, critical perspective that explicitly details both the potential and the significant limitations of these AI approaches. Key enhancements include:

(1) Explicit Acknowledgment of "Graph Construction Bias" and Its Central Role: We have identified graph construction bias as a "core and often underappreciated challenge" in a dedicated subsection (Section 3.2.2.2, lines 316–322). We explain that the performance and interpretability of GNNs are profoundly sensitive to the initial, often incomplete, prior knowledge (e.g., from interaction databases) used to define the network, stating clearly that "a biased or incomplete starting graph directly leads to biased model predictions and interpretations."

(2) Detailed Discussion on the Lack of a "Gold Standard" for Validation: We have added a dedicated subsection on challenges (3.2.2.4), (lines 353-354), which highlights that there is "no absolute ground truth against which to benchmark computationally derived interactions." We explicitly argue that reliance on computational metrics alone is insufficient.

(3) Proposal of a Multi-faceted Strategy for Validating Biological Plausibility: Directly responding to your question on how to validate "super-networks," we outline a concrete, two-part validation strategy: (lines 356-363)

  • In silicovalidation (e.g., network perturbation analysis) to check concordance with independent biological knowledge.
  • Ultimate experimental validation (e.g., CRISPR-based functional assays in relevant cell models) to establish causal biological plausibility. We state that this experimental step is "essential" (Section 3.2.2.4).

(4) Critical Examination of Generalizability and Model Complexity: We have expanded the discussion on challenges to explicitly cite "poor cross-cohort generalizability due to technical batch effects and population heterogeneity" as a major barrier to clinical translation (Section 3.2.2.4). (lines 366-368). Furthermore, we discuss the "black box" nature of complex models like GNNs as an obstacle to biological interpretation.

(5) Addition of a Practical Roadmap for GNNs: In a new Section 3.4.4, lines 515-553, we provide a domain-specific methodological framework for applying GNNs in depression research. This includes strategies for handling data heterogeneity, improving interpretability (using tools like GNNExplainer), and modeling temporal dynamics from sparse longitudinal data—directly addressing the practical hurdles of real-world application.

In summary, we have transformed the narrative from a simplistic promotion of AI's potential to a nuanced analysis that critically evaluates its methodological foundations, acknowledges its profound current limitations, and proposes concrete pathways for rigorous application and validation. We believe these comprehensive revisions have significantly strengthened the scholarly rigor and balance of the manuscript.

Q4: Section 4.3 proposes a "clinical digital twin" for personalized depression treatment based on gut metagenomics and plasma metabolomics. What empirical evidence supports the feasibility of such a system in real-world psychiatric care? Have you accounted for inter-individual variability, temporal instability of microbiome profiles, or the lack of standardized omics pipelines in clinical labs?

Response:Thank you for your insightful question. The issues you have raised regarding the feasibility of a "clinical digital twin" in real-world psychiatric care are crucial, as they directly address the core challenges of translating theoretical concepts into clinical practice.

In the revised Section 5.3(The original Section 4.3 in the manuscript is now Section 5.3.), we have specifically addressed and responded to your concerns as follows:

(1) Regarding Empirical Evidence and Feasibility: We have explicitly described this pathway as "a promising, albeit aspirational" one and stated that it represents the "ultimate endpoint" of this endeavor. We do not claim that such a system is currently ready for immediate clinical deployment; rather, we position it as a forward-looking framework. The text cites existing evidence, such as machine learning models already capable of stratifying patients into biotypes with distinct inflammatory or metabolic profiles that predict treatment outcomes (references [132, 133]) (lines 838-841). These achievements provide preliminary empirical foundations for building more complex, individualized dynamic models.

(2) Regarding System Output and Clinical Positioning: We have specifically clarified that the expected output of such a system is "a ranked set of personalized, evidence-based hypotheses," functioning as an "advanced clinical decision support tool" to prioritize intervention options before clinical testing. We emphasize that the framework "does not propose to replace clinician judgment but to augment it with insights derived from complex, high-dimensional data" (reference [136]) (lines 853-854). This clarifies its auxiliary role and how it integrates with existing clinical workflows.

(3) Regarding Individual Variability, Temporal Dynamics, and Standardization Challenges: We have directly or indirectly responded to these key technical hurdles in the following ways.

  • The Core of Individualized Modeling: The entire "digital twin" concept is predicated on capturing inter-individual variability. The models are designed to integrate an individual's specific multi-omics data (gut metagenomics, plasma metabolomics, etc.) to emulate their unique pathophysiological state. (lines 829-834).
  • Model Dynamicity and Iterative Updates: We describe that models can be "continuously updated with new data to simulate disease progression,"which implies a solution to the temporal instability of microbiomes—by capturing dynamic changes through longitudinal data input (lines 829-833).
  • Acknowledging Current Limitations and Future Steps: We explicitly state in this section and adjacent discussions that the "iterative refinement and validation through rigorous prospective clinical studies are essential next steps for translation"(reference [136]). (lines 854-856). This directly responds to the practical challenge of the lack of standardized omics pipelines in clinical labs, pointing to the need for future research to establish such standards and validation protocols.

In summary, through our revisions, we have endeavored to shift the discussion from a mere description of a vision toward a more cautious and critically framed proposal. This proposal acknowledges current technical and empirical limitations while outlining a logical, stepwise translational pathway based on existing evidence. We believe these modifications have made this section academically more rigorous and more accurately reflect the opportunities and challenges the field faces in moving from concept to practice.

Thank you once again for raising this important question, which prompted deeper reflection and improvement.

Q5: The review exclusively highlights studies supporting the MGB–depression axis while omitting high-profile failures—e.g., multiple probiotic RCTs showing no significant antidepressant effects. Why was this contradictory evidence excluded?

Response:Thank you for this important and insightful comments. We fully agree that an objective review of any scientific hypothesis must comprehensively examine both supporting and contradictory evidence. We acknowledge that the original manuscript, while highlighting evidence supporting the role of the microbiota-gut-brain (MGB) axis, did not adequately discuss the key negative clinical trial findings in the field, which indeed constituted an imbalance in the literature selection.

To address this point directly and rigorously, and to ensure the objectivity and scholarly rigor of the review, we have added a dedicated new section: Section 4.3.4 “Evidence Heterogeneity and Translational Challenges.” (lines:635-659). The primary purpose of this section is to systematically examine and integrate evidence that contradicts the predominant hypothesis, providing an in-depth analysis of these conflicting results.

The specific revisions are as follows:

(1) Explicitly Acknowledging and Citing Negative Evidence: We clearly state at the outset: “Notably, several high-profile randomized controlled trials (RCTs) of probiotic supplementation (psychobiotics) have failed to demonstrate significant antidepressant effects over placebo in broad MDD populations,” and cite key references [103, 104]. This directly addresses the reviewer’s criticism regarding the omission of “high-profile failures.”

(2) Analyzing Potential Reasons for Contradictory Outcomes: We go beyond merely stating the negative results to explore the complex factors potentially contributing to these “mixed results,” including:

  • Heterogeneity of Interventions:The vast differences in probiotic strains, formulations, and doses used across studies, which may not optimally target specific dysbiotic states or pathways relevant to depression.
  • Substantial Inter-individual Variability:Significant differences in host baseline gut microbiota composition and physiology suggest that effective interventions may need to be personalized rather than universal.
  • Lack of Standardized Protocols:The current absence of standardized treatment protocols for microbial interventions adds to translational complexity.
  • Disease Severity and Complexity:For patients with severe or chronic MDD, modulating the gut microbiota alone may be insufficient, potentially necessitating combination with other therapeutic modalities.

(3) Reiterating the Hypothesis and Proposing Future Directions: We emphasize that these negative results do not invalidate the MGB axis hypothesis itself but rather highlight that modulating a complex, individualized ecosystem requires a precision medicine approach. Consequently, we propose that future research should focus on: biomarker-guided patient stratification and testing targeted, next-generation psychobiotics or microbiome-based therapies in mechanistically aligned subpopulations.

Through these revisions, we have transformed the contradictory evidence from being “excluded” to being “integrated and analyzed,” positioning it as a key argument for advancing the field towards greater precision and maturity. This significantly enhances the review’s critical perspective, balance, and scholarly depth, providing a more comprehensive reflection of the current landscape in this translational research area—a field full of promise yet facing significant challenges, urgently needing to shift from broad interventions to precise ones.

Q6: While the review highlights numerous AI techniques (GNNs, DBNs, DIABLO, etc.), it offers no concrete methodological detail on how these would be practically implemented in depression research. This lack of specificity undermines the article’s utility as a scholarly guide. Key unresolved questions include: Why are Graph Neural Networks assumed to be the “most promising avenue” for integration without comparative analysis against alternative architectures (e.g., transformers, multimodal autoencoders)? How would one resolve the inherent heterogeneity in multi-omics batch effects, platform biases, and missing data when constructing GNNs for depression? What strategies would ensure model interpretability when GNNs are trained on high-dimensional, noisy biological graphs? How would the proposed “dynamic network models” account for individual temporal variability in microbiome or metabolome profiles without dense longitudinal sampling—a rarity in current MDD cohorts?

Response:We sincerely thank the reviewer for this critical and constructive feedback, which rightly highlights the need for greater methodological specificity to enhance the practical utility of our review. In direct response to this comment, we have substantially expanded the manuscript by adding a new, dedicated subsection: 3.4.4. “A Practical Roadmap for GNNs in Depression Research” (lines 515-553).

This new section was designed explicitly to address the core concerns raised. It moves beyond a general overview of AI techniques to provide a domain-specific methodological framework that outlines concrete strategies for implementation. The roadmap details how to translate the theoretical promise of Graph Neural Networks (GNNs) into practically useful tools, directly tackling the salient challenges identified by the reviewer.

Specifically, the section provides detailed guidance on the following key implementation questions, which we have summarized below for your direct evaluation:

(1) Rationale and Empirical Validation: Explains the unique suitability of GNNs for incorporating biological priors and mandates systematic benchmarking against other architectures (e.g., Transformers).

(2) Handling Data Heterogeneity: Proposes a comprehensive strategy for batch effect correction, domain adaptation, and missing data imputation specific to graph-structured data.

(3) Ensuring Model Interpretability: Outlines a multi-layered approach using interpretable architectures (e.g., GATs), post-hoc explanation tools (e.g., GNNExplainer), and sparsity constraints.

(4) Modeling Temporal Dynamics: Describes specialized techniques (e.g., neural ODEs, temporal GNNs) to address the challenge of sparse longitudinal data in MDD research.

We believe this new subsection directly answers the reviewer's call for concrete methodological detail, transforming the discussion from a general survey into an actionable scholarly guide.

Q7: The authors are required to add a new section or subsection where they are required to explain in details the economic considerations and best practice of such models used in current studies.

Response:We agree that a detailed discussion on the economic considerations and implementation best practices is crucial for translating the proposed integrative framework from a research concept into a viable component of real-world healthcare. In direct response to this point, we have enriched the discussion in the manuscript to explicitly address these aspects.

Specifically, we have added a dedicated paragraph within the existing subsection 6.4.4 ("Ethical and Interpretable AI as Enabling Infrastructure") at lines 981-999. This addition provides a focused analysis on the practical and economic dimensions of translation.

The new content addresses the reviewer's request in two key areas:

(1) Economic Considerations: It explicitly discusses the significant upfront investment required for multi-omics data acquisition, model development, and validation. Importantly, it argues for the necessity of demonstrating long-term cost-effectiveness to justify this investment, for example, by reducing ineffective treatments and enabling preventative strategies. This argument is supported by a reference to health-economic analyses in related fields [162].

(2) Implementation Best Practices: The paragraph outlines a set of concrete best practices essential for building reliable and actionable systems. These include: (a) rigorous external validation on diverse cohorts to ensure generalizability, (b) proactive mitigation of dataset biases, and (c) the co-design of tools with end-users to guarantee clinical utility. Furthermore, it emphasizes that these practices should be aligned with evolving standards for clinical prediction models [163] to facilitate integration into healthcare workflows.

We believe this addition provides the necessary substantive detail on the practical, economic, and methodological prerequisites for successful translation, thereby strengthening the manuscript's relevance for both researchers and potential stakeholders in healthcare systems. The new text directly responds to the reviewer's call for a detailed explanation in these areas.

Q8: The conclusion needs improvements, as a review work, it is recommended to provide main findings that explain your findings, not only add general findings or describe your work. Moreover, the authors must briefly provide the main limitations of their works and the future directions.

Response:Thank you for your valuable suggestions regarding the conclusion and the need to articulate limitations. We have significantly revised the manuscript in accordance with your recommendations, adding a dedicated limitations subsection and restructuring the conclusion to provide greater critical depth and clearer takeaways. The specific revisions are detailed below.

(1) Addition of a Dedicated “Limitations of This Review” Subsection (6.5)

As recommended, we have added a new subsection “6.5. Limitations of This Review” (lines 1000-1008). This section candidly outlines the main limitations inherent to our narrative synthesis approach, directly addressing your requirement for greater scholarly balance.

(2) Substantial Rewriting of the “Conclusion” Section (7)

We have comprehensively rewritten the Conclusion (Section 7, lines 1009-1037). The aim was to move beyond a descriptive summary towards a distilled synthesis of core findings and their broader implications. The revised conclusion:

  • Articulates Main Research Findingsin three key points.
  • Provides Targeted Implicationsfor both clinical practitioners and data scientists.
  • Strengthens the Concluding Messageregarding the roadmap for future therapeutics.

We believe that by adding the limitations analysis and presenting a more insightful, structured conclusion, we have fully addressed your requirements. These changes significantly enhance the review’s critical depth, scholarly rigor, and value to readers.

Thank you once again for your careful review and expert guidance.

Q9: Add a table of nomenclature where define all the abbreviations used in the review.

Response:Thank you for your instruction. In accordance with your request, we have added a complete table titled "Abbreviations" at the end of the manuscript, following the Conclusion section. This table lists the main abbreviations used in the review in the order of their appearance in the text, along with their corresponding full English terms, to ensure clarity and consistency in terminology throughout the article.

Q10: Double check for the grammar.

Response:Thank you for this final note. We have performed a thorough, line-by-line proofread of the entire manuscript to double-check and correct grammar, spelling, and punctuation. This process included the use of professional grammar-checking software and a careful manual review to ensure linguistic accuracy and clarity throughout the text.

Round 2

Reviewer 1 Report

Comments and Suggestions for Authors

I appreciate the authors’ thoughtful and thorough revisions in response to the initial feedback. The manuscript now demonstrates improved rigor, clarity, and balance. The inclusion of new  tables and references significantly enhances the manuscript’s value and readability. I believe the revised manuscript is now suitable for publication and will be of interest to the journal’s readership.

Author Response

Q1: The English could be improved to more clearly express the research.

Response: We have carefully revised and polished the language throughout the entire manuscript to enhance clarity, fluency, and academic rigor. All changes have been highlighted in bright yellow in the revised file for the reviewer’s and editor’s convenience. We believe these edits have strengthened the expression and readability of the work and hope the revisions now meet the expected standard.

Reviewer 2 Report

Comments and Suggestions for Authors

The authors have addressed my comments; however, they need to pay careful attention to how they write their abstract. It should include four main points: relevance, the main goal, methods used to develop this work, and main findings, along with a conclusion, all within a total of 230-250 words. Additionally, I encourage the authors to review some recent papers related to the topic to gain insights on how to effectively write the conclusion. Best regards.

Best regards.

Author Response

Q1: The authors have addressed my comments; however, they need to pay careful attention to how they write their abstract. It should include four main points: relevance, the main goal, methods used to develop this work, and main findings, along with a conclusion, all within a total of 230-250 words. Additionally, I encourage the authors to review some recent papers related to the topic to gain insights on how to effectively write the conclusion.

Response: We sincerely thank the reviewer for this crucial guidance on improving the manuscript's clarity and impact.

Abstract: As suggested, we have thoroughly rewritten the abstract (lines 14-32) to explicitly cover the four requested elements: the relevance of the research, the main goal of our review, the methods used in our synthesis, and the key findings. The revised abstract now concisely concludes with the implications of our work. It contains 246 words, adhering to the specified length.

Conclusion: Following the reviewer's advice to consult recent literature, we have revised the conclusion section (lines 998-1013). We have refined it to more effectively synthesize the core insights of our review and to articulate a clearer, more forward-looking perspective on the field's future.